# Using Low-Cost Radar Sensors and Action Cameras to Measure Inter-Vehicle Distances in Real-World Truck Platooning

**Markus Metallinos Log [1,\*]**, **Thomas Thoresen [2]**, **Maren H. R. Eitrheim [1,3]**, **Tomas Levin [4]** and **Trude Tørset [1]**

1   Department of Civil and Environmental Engineering, Norwegian University of Science and Technology (NTNU), NO-7491 Trondheim, Norway
2   Research Division Strategic Analyses and Joint Systems, Norwegian Defence Research Establishment (FFI), NO-2027 Kjeller, Norway
3   Department of Humans and Automation, Institute for Energy Technology (IFE), NO-1751 Halden, Norway
4   Norwegian Public Roads Administration (NPRA), NO-7031 Trondheim, Norway
\*   Correspondence: markus.log@ntnu.no

**Abstract:** Many modern vehicles collect inter-vehicle distance data from radar sensors as input to driver assistance systems. However, vehicle manufacturers often use proprietary algorithms to conceal the collected data, making them inaccessible to external individuals, such as researchers. Aftermarket sensors may circumvent this issue. This study investigated the use of low-cost radar sensors to determine inter-vehicle distances during real-world semi-automated truck platooning on two-way, two-lane rural roads. Radar data from the two follower trucks in a three-truck platoon were collected, synchronized and filtered. The sensors measured distance, relative velocity and signal-to-noise ratio. Dashboard camera footage was collected, coded and synchronized to the radar data, providing context about the driving situation, such as oncoming trucks, roundabouts and tunnels. The sensors had different configuration parameters, suggested by the supplier, to avoid signal interference. With parameters as chosen, sensor ranges, inferred from maximum distance measurements, were approximately 74 and 71 m. These values were almost on par with theoretical calculations. The sensors captured the preceding truck for 83–85% of the time where they had the preceding truck within range, and 95–96% of the time in tunnels. While roundabouts are problematic, the sensors are feasible for collecting inter-vehicle distance data during truck platooning.

**Keywords:** inter-vehicle distance measurements; radar sensor; action camera; field study; automated truck platooning; rural road; manual video coding; field-of-view; tunnels; roundabouts

## 1. Introduction

Truck platooning refers to the innovative concept of wirelessly linking trucks into convoys using adaptive cruise control (ACC). Platoons consist of one lead truck and one or more following trucks. Wireless communication may enable shorter inter-vehicle distances than those currently considered safe for manually driven trucks, which are constrained by the reaction times of human drivers [1,2]. Shorter inter-vehicle distances lead to a reduction in aerodynamic drag, which may enable fuel savings and reduced emissions. Moreover, tight, automated vehicle control may unlock improvements in safety and efficiency of the road traffic system [2]. Platooning may also benefit society at large, in terms of cheaper, safer and more streamlined road freight operations. However, platooning is yet to be commercially deployed and there are many unanswered questions.

Field studies are often used to explore the technology and are organized either by transport companies [3] or truck manufacturers directly [4], or through larger, publicly funded undertakings, such as the KONVOI [1] and ENSEMBLE projects [5]. The studies have typically been conducted on highways. Few studies, if any, have investigated platooning on challenging two-way, two-lane roads with oncoming traffic and narrow tunnels. In Norway, for example, large parts of the road network are subject to such issues,

on which more research is needed to establish the feasibility of truck platooning. Public roads authorities govern the design and operations of the road network and may thus be important facilitators of truck platooning. As Norway is a small automotive market with conditions for automated vehicles, the Norwegian Public Roads Administration (NPRA) have taken a proactive role in trialing advanced transportation technologies, exemplified by the Borealis testbed [6] and the ongoing MODI project [7], which aims to demonstrate automated trucking between the Netherlands and Norway within 2026.

In general, the shorter inter-vehicle distances are between platooning trucks, the greater are the resulting benefits in terms of fuel savings and potential road capacity improvements. Thus, the greater the stability of inter-vehicle distances over time, the more beneficial platooning will be [8]. However, during real-world driving, combinations of external traffic, road alignment and truck weight differences will influence the inter-vehicle distances, and thus also the extent of benefits unlocked [9,10]. Moreover, the extent to which truck platoons impact surrounding traffic depends partly on their total length, which is influenced by inter-vehicle distances. By implication, inter-vehicle distance data will be important for public roads authorities when regulating truck platooning, such as when deciding on which road sections platoons should be allowed, and the maximum number of trucks which can platoon together. However, such data may not be easily accessible, and even if they were, it is unclear how they should be contextualized and analyzed.

In the field studies organized through the large, aforementioned platooning projects, truck manufacturers and transport companies have typically facilitated and allowed for the collection of inter-vehicle distance measurements using integrated vehicle sensors. While integrated distance sensors could also be cameras and lidars [11], radar sensors are most often used, as they are affordable, computationally simple and robust under adverse light and weather conditions [11–15]. This is despite issues with clutter [16] and ghosting [17], referring to unwanted signals that distort and interfere with the desired detection, and the detection of non-existent targets which are difficult to distinguish from real ones.

If allowed, using data from integrated vehicle sensors is very convenient, as they are already collected as real-time inputs to platooning control systems, removing the need for using aftermarket distance sensors. However, manufacturers often use proprietary algorithms for data processing, defining message codes [18] to encrypt collected data. This makes external individuals unable to access them, unless authorized to do so. Smaller truck platooning field studies may not have the benefit of truck manufacturers participating as partners. Some may also seek to verify data from manufacturers, for which independent methods for collecting such data would be useful.

High-precision global navigation satellite system (GNSS) receivers located in successive vehicles could theoretically be used for this purpose [19,20]. Some areas, however, have road tunnels and topographical features where GNSS-based data collection methods would be subject to signal blockages [12,20]. External aftermarket radar sensors may circumvent the problem, provided they are adequately practical and accurate. Truck platooning research is often publicly funded, so solutions should preferably be low-cost. Since the output of aftermarket radar sensors would not be used for operative vehicle control, they do not need to be as capable, nor provide the same level of reliability as automotive-grade distance sensors, both of which drive cost and complexity. They may also be more flexible, in terms of allowing for custom placement and user adaptations.

Many researchers have focused on perception, functional safety and operative control for truck platoons [21]. These studies often include cameras and radars. However, for studying the effects of truck platooning from the standpoint of roads authorities, these methods are more computationally complex than they need to be, and simple methods for estimating inter-vehicle distances from truck platooning would be useful.

The current study investigated the feasibility of using Anteral universal radar (uRAD) sensors for Raspberry Pi [22] to measure inter-vehicle distances in a truck platoon on rural roads. This application represents a novel use case for this type of sensor. We propose a multi-faceted approach for collecting inter-vehicle distance data from truck

platooning field trials. It aims to provide technical details and best-practices on data collection, synchronization, filtering and analysis. Dashboard cameras in each truck filmed the driving scene. This footage was used to log the timestamps of specific, recurring events, allowing for exploring sensor operation in different driving conditions.

The paper addresses the research question: How can low-cost radar sensors and action cameras be used to investigate inter-vehicle distances in real-world truck platooning?

## 2. Materials and Methods

This section provides details on the data collection set-up, the equipment used, and the procedures for synchronization, video coding and radar data processing.

### 2.1. Data Collection Set-Up

A truck platooning field trial was undertaken on public rural roads in northern Norway in the fall of 2020. This is the first study of its kind, and was also reported in [23]. Three drivers operated three semi-trailer trucks along a 380 km two-way, two-lane road stretch traversing a mountainous, coastal area. The trucks were numbered 1, 2 and 3, based on the main truck order configuration. One section was traversed repeatedly with different orders. A prototype ACC system was installed, enabling the trucks to operate as a platoon when detecting a preceding truck. Data from integrated cameras and radars were unavailable. Aftermarket equipment was used to collect data over 7 h of driving.

While longitudinal control was automated, the drivers operated the wheel manually, placing the field study at the Society of Automotive Engineers (SAE) Level 1 [24]. All trucks had 500 horsepower. Trucks 1 and 2 had equal weights, while truck 3 was lighter. (41 and 27.5 metric tons), reflecting the scenario of trucks encountering each other to platoon during real-world operations, unharmonized with respect to weight. Consequently, inter-vehicle distances fluctuated, as the platoon often struggled to remain collected on the winding road. Twenty-three tunnels and eleven roundabouts were traversed. The most prevalent speed limits were 80 and 60 km/h, during which inter-vehicle distances at 3-s time gaps were 50–70 m, i.e., comparable to manual driving. Figure 1 illustrates the set-up.

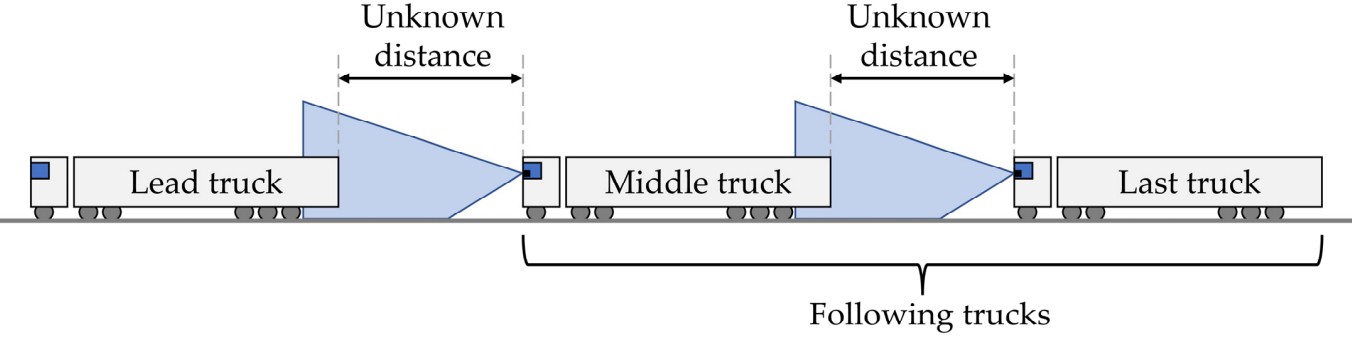

**Figure 1.** Truck platooning set-up. Radar sensors were mounted on windshields of following trucks.

Herein, *preceding truck* is a term used for the truck located in front of the truck in question. Depending on the context, it may refer to either the lead truck or the middle truck. In truck order 1-2-3, truck 1 precedes truck 2, and truck 2 precedes truck 3. The terms *leader* or *leading truck* are only used for the truck located at the front of the platoon, while *followers* refer to both the middle and last truck together.

The two rearmost trucks were identically instrumented. The Raspberry Pi, with the uRAD sensor attached, was fixed to the inside of the windshield using a suction mount with a flexible arm. The mount did not interfere with the field-of-view of the sensors. A portable monitor was used to administer radar logging. Sensors were placed at slightly different heights in each truck due to interior constraints, cf. Appendix A.1.

Two GoPro video cameras were also mounted: A windshield-mounted dashboard camera filmed the driving scene, and another camera filmed the interior. Footage from

the latter was only used for synchronizing radar data to the dashboard footage. The study was approved by the Norwegian Centre for Research Data (457013). Participants agreed to being recorded, and all videos and audio were handled and stored confidentially.

The equipment in each truck was started and stopped in succession when trucks were parked, but was left on during short breaks. None of the Raspberry Pi microprocessors had internet connection, so they did not adhere to local time. An equipment start-up procedure was devised which allowed for post-hoc synchronization of videos and radar data. For each truck, all cameras were started before starting radar logging. When starting each camera, the Emerald Sequoia Time smartphone application was presented. Using Network Time Protocol (NTP) servers, which synchronizes computer clocks over the internet [25], this application provides more accurate times than those typically provided by internal clocks [26]. The application shows local time, and, when cellular reception is available, it calculates deviations from NTP time. The mean offset was 0.08 s, i.e., negligible.

GNSS data were collected from VBOX Sport loggers and a fleet management system (FMS), in an effort to compute inter-vehicle distances to validate the radar data. A script was written to interpolate timestamps and calculate distances between GNSS locations from each truck. The loggers were supposed to activate automatically [27], but this functionality occasionally failed. GNSS files were also extracted from the FMS, and all files were visualized in QGIS. Both systems experienced outages in tunnels. While the FMS had good positioning accuracy outside tunnels, its update rate was too low, and loggings were not always synchronized across the trucks. VBOX data which did get collected had frequent outages, and timestamps were often erroneous, placing trucks in incorrect order. This highlights the utility of radar in estimating inter-vehicle distances in such areas.

### 2.2. Radar Sensors

Frequency modulated continuous waves (FMCWs) are radar waveforms often used to measure distances in automotive applications [13,28]. Anteral uRAD radar sensors for Raspberry Pi were tested here, shown in Figure 2. These are 24 GHz FMCW radar sensors which connect conveniently as extension boards to Raspberry Pi microprocessors [29]. Such microprocessors run a user-friendly operating system and can interface with purpose-built components. The automotive industry is increasingly using 77 GHz radar sensors, allowing for increased range resolution and accuracy [30]. These sensors can better separate closely spaced objects, and can be packaged in a smaller form factor. However, 24 GHz sensors are less expensive, and automotive-grade 77 GHz sensors which could be operated from Raspberry Pi microprocessors were not available when procuring the equipment. Thus, testing the proposed methodology using cheaper 24 GHz sensors was considered reasonable. While many 77 GHz FMCW radar sensors are more range-capable and have wider fields-of view, some 77 GHz radars, e.g., in [31], have shorter ranges than the uRAD sensors.

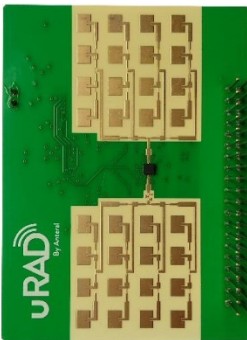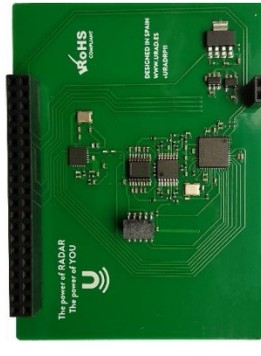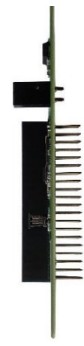

**Figure 2.** Radar sensor outline [22].

The uRAD sensors have a theoretical distance range approaching 100 m, 30° fields-of-view both horizontally and vertically, and are able to detect up to five objects simultaneously. For each object, distance, radial velocity and reflected power (SNR, i.e., signal-to-noise ratio) are registered. In the context of truck platooning, both the sensor and the desired objects are in motion; so *relative* velocities are detected. The velocity range (±0.2 to 75 m per second) is within the range of values encountered in road traffic.

The use case and intended sensor placement were described to the supplier, which expected the application to be feasible, but did note that similar tests had never been carried out before. The sensor has been used for other purposes, the most relevant of which is as a stationary speed sensor [29,32,33]. However, transferability from the cited studies is limited, due to different use cases and configurations.

Eight parameters were used to configure the sensors. These are detailed in Appendix A.1, alongside pre-trial testing of the mounting set-up and parameters. The most important considerations are detailed here.

Firstly, the radar mode details the waveforms transmitted by the sensor. Triangular waves were chosen, which maximized the sensor range and the update rate for outputs (9–13 Hz) from the radar script. This mode also allowed for subsequent data filtering based on relative velocity. Maximizing range was important for capturing data even when trucks were located far apart, as the trucks were expected to drive with human-level gap sizes (2–3 s) at distances approaching the upper distance range. Adverse road geometry would presumably also lead to safer driving at larger gap sizes. Maximizing the update rate was seen as beneficial for obtaining as many measurements as possible. The update rate of the uRAD radar sensor is comparable to the 77 GHz sensors showcased in [14].

Secondly, number of targets detected ($N_{tar}$) and the detection distance ($R_{max}$) were maximized, to capture the most data, and to enable filtering of unwanted detections later.

Thirdly, moving Target Indicator (*MTI*) was activated, for including data only from objects with motion relative to the sensor. The supplier stated that it would only eliminate objects which were absolutely static, such as detections of the windshield. The preceding truck would still be registered, even when moving at the same velocity as the sensor.

Fourthly, for each truck, different values for ramp start frequency ($f_0$) and the duration of each wave ramp ($N_s$) were used for each sensor to avoid interference. Since each sensor had different $N_s$ and $f_0$ values, their theoretical maximum distance ranges also differed, at 75.0 and 73.1 m, for trucks 2 and 3, respectively, based on Equation A3 in Appendix A.1. These values are in line with 70–75 m estimates from the supplier. For comparison, automotive radar sensors typically have ranges of 30–150 m [30,34]. The sensors had a stated distance accuracy of ±0.3%, corresponding to a ±0.23 m deviation at 75 m, which is considered sufficient for the current use case. Table A3 in Appendix A.1 shows all parameters which were used.

### 2.3. Video Footage, Synchronization and Manual Video Coding

Dashboard footage was recorded for exploring the radar data as a function of the driving scene. Without the videos, this would not have been possible. First, footage had to be coded, i.e., timestamps had to be established for relevant events in the footage. This is different from the more computationally complex process of semantic segmentation used in computer vision, which involves categorizing relevant objects in the scene, often using bounding boxes [31].

Video footage was synchronized and aligned to local time using BORIS, i.e., Behavioral Observation Research Interactive Software [35]. BORIS is a free, open-source video coding program. Each BORIS observation contained videos from the same truck and driving stretch. By checking time differences as displayed by the phone application at the start of each recording, time offsets were established, achieving near-perfect synchronization. The date and time of each observation was defined as the local time shown by the phone application to the longest video file, as the recording was started. This ensured that all videos from the trucks were aligned to local time during the field trial.

Events in BORIS were defined using the ethnogram, and were either point or state events: Point events had no duration (i.e., having only one timestamp), while state events did (i.e., having both start and end timestamps). Video coding was carried out while playing videos at 2–4 times normal speed, depending on driving scene complexity. Events were coded by the first author, ensuring consistency. Videos codes were subsequently reviewed by the third author. Onwards, *italics* are used to refer to the video codes, as illustrated in Figure 3. Radar initiations (*Radar logging*) and oncoming traffic were defined as point events. Amongst oncoming vehicles, only *Trucks* seemed to affect inter-vehicle distances during platooning. Video footage showed that when encountering large trucks on narrow road segments, the lead truck often reduced its speed, causing speed reductions also for the followers and a contraction in inter-vehicle distances. Truck order codes indicate which periods the instrumented trucks collected relevant inter-vehicle distance data within the platoon, as opposed to the periods where they served as platoon leaders, collecting irrelevant data preceding the platoon.

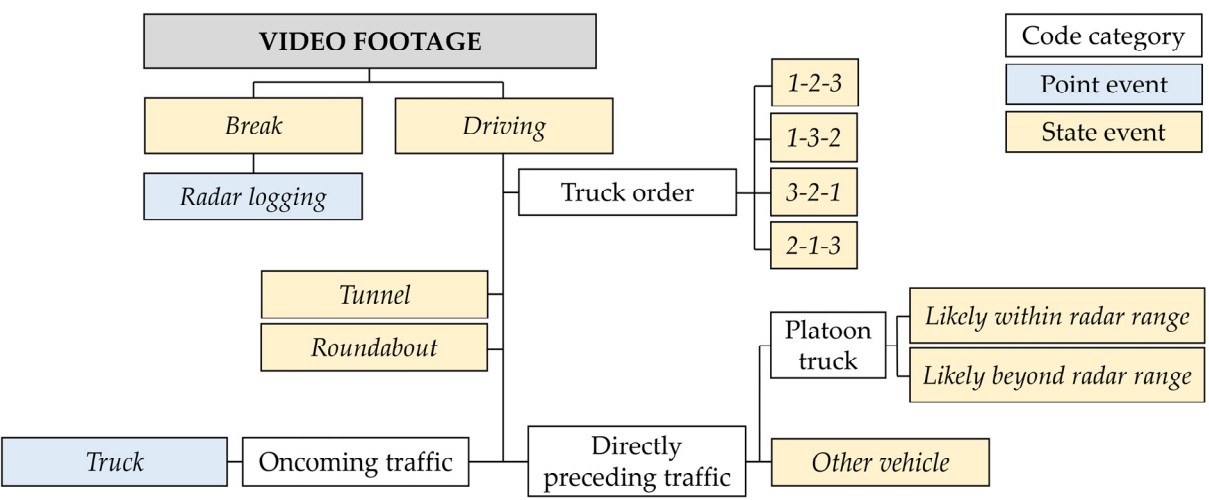

**Figure 3.** Overview of BORIS video codes.

The *Driving* video code in Figure 3 includes all conditions encountered, i.e., including tunnels and roundabouts, thus showcasing diverse, complex driving segments. See Table A5 in Appendix A.2 for its definition. *Tunnel* and *Roundabout* video codes denote scenarios of particular interest. In tunnels, inter-vehicle distances between the trucks cannot be determined using GNSS-based methods, and it is also unclear how tunnels affect operating conditions for the radar sensors. Roundabouts are demarcated areas (i.e., they are simple to code from video footage) with small horizontal radii, which can illustrate effects of road curvature on radar operation when the preceding truck turns. All events were coded separately for each truck. Tunnels were coded from the moment when the front of the truck in question entered the tunnel, to when the front of the truck left the tunnel. The same principle was used for roundabouts, i.e., coding the moment when the front of each truck entered and exited the circulating area.

Events were defined for visual inspection of the distance to (visibility of) the preceding truck, as the trucks were at times located far apart. The goal was to remove data from periods when the preceding truck was difficult or impossible for the sensor to detect, due to the driving situation. This occurred in two scenarios. Firstly, it occurred in sharp turns, where the preceding truck would disappear from radar field-of-view. Dashboard cameras had larger horizontal fields-of-view than radar sensors, so when no preceding vehicle appeared on camera, the radar would also not detect it. Secondly, it occurred when the trucks drove far apart. The distance range was shorter for radar than for the dashboard camera, which was only constrained by line-of-sight. Both scenarios were coded as *Likely*

*beyond radar range (LBRR)*. Conversely, *Likely within radar range (LWRR)* denotes driving periods when relevant radar data likely could have been collected.

The manual nature of this process introduces some limitations. Periods when the preceding truck was actually *LWRR* may have been coded as *LBRR*, and vice versa. Transitions between these codes may also occur at different distances. Albeit imperfect, this categorization is preferable versus including all radar data, even when the preceding truck was located far beyond radar range. For the far-apart scenario, centerline road markings initially aided the visual estimation. On rural roads with 80–90 km/h speed limits, these markings have standardized lengths and gaps totaling 12 m, which repeat continuously, cf. pp. 22 in the Norwegian road marking design manual [36]. After having coded *LWRR* and *LBRR* using road markings for some time, the remaining dashboard camera footage was coded without conscious reference to the road markings. It was also attempted to use pixel counts of the preceding truck for this purpose, but doing so at large scale was unsuccessful. An overview of the data collection and processing steps is provided in Table 1. See Table A5 in Appendix A.2 for examples of video codes.

**Table 1.** Overview of the data collection and processing steps.

| Step | Context | Description |
|---|---|---|
| 1 | Equipment start-up and logging | Start GoPro-cameras successively, while, for each camera, presenting local time on phone screen. |
| 2 | | Start radar logging script while producing loud verbal cue. |
| 3 | Data collection | Platoon driving. |
| 4 | Equipment logging stop | Stop GoPro camera recordings successively. Stop radar logging. |
| 5 | Data transfer | Import GoPro video files and raw radar files to computer. |
| 6 | Synchronize GoPro videos with each other | For each truck: Synchronize GoPro video footage in BORIS, using offset values. Synchronization is based on the difference between local time presented to each camera upon starting the recordings, and fine-tuned using recorded audio. |
| 7 | Synchronize GoPro videos to local time | Define *Date and time* in BORIS observation equal to the local time shown to the reference camera (i.e., the longest video file) by the phone application when the reference recording was started. |
| 8 | Video coding | Code *Radar logging* based on visual and verbal cues from interior camera. Code remaining events from dashboard camera footage. |
| 9 | Synchronize radar data to local time | Export events list for each observation to spreadsheets. |
| 10 | | Apply datetime shift to radar timestamps based on *Date and time* for each BORIS observation to match them with *Radar logging* events. |
| 11 | Radar data curation | Radar data were curated using six filters. |

### 2.4. Radar Data Processing

Video coding events were exported from BORIS as spreadsheets, and the *Date and time* from the corresponding *observation* was added to the timestamp of each instance of the *Radar logging* video code. This assigned local time to the instance when radar logging was started, and served as basis for synchronizing video codes and radar data in Python.

Filters were needed to extract only the inter-vehicle distances between the platooning trucks. Filtering aimed at removing data from periods when the trucks were not driving (i), data which did not correspond to the preceding truck (ii), noise (iii), and finally, data from periods when the preceding truck was outside sensor range (iv).

Timestamps, distance (m), relative velocity (km/h) and signal-to-noise ratio (decibel, dB) were logged, for up to five simultaneously detected targets. Positive relative velocities corresponded to targets receding from the radar, and negative relative velocities corresponded to approaching targets. For the curated radar dataset, an example of the former would be the preceding truck accelerating away from the truck in question. Conversely, the preceding truck decelerating would be an example of the latter. SNR denotes the ratio of the signal power to the noise power. Larger and more reflective objects will produce measurements with higher SNR values. The radar data were curated using successive filters, cf. Table 2. The following paragraph outlines the details and purpose of each filter.

**Table 2.** Overview of radar data filters.

| Filter | Description |
| --- | --- |
| 1 | *Driving* and *following* |
| 2 | Relative velocity within ±30 km/h |
| 3 | Signal-to-noise ratio < 15 dB |
| 4 | Target selection |
| 5 | Downsampling 1 Hz |
| 6 | *Likely within radar range* (*LWRR*) |

First, the *Driving* video code was used to remove data collected during irrelevant periods. It discards data from *Break* periods, so only data from *Driving* periods remain. Simultaneously, truck order codes were used to exclude radar data collected during periods when each truck served as platoon leader. Specifically, radar data from truck 2 stem from the driving periods with truck orders *1–2–3*, *1–3–2* and *3–2–1*, while radar data from truck 3 stem from periods coded as *1–2–3*, *1–3–2* and *2–1–3*. Data were discarded from periods where external vehicles (*Other vehicle*) preceded each respective truck.

Filters for relative velocity and SNR (filters 2 and 3, respectively), were used to clean the remaining data. Relative velocities were explored in histograms, shown in Figure 4a. Some relative velocity bins were far more frequent than others, giving the dataset large dynamic range. The vertical axis is logarithmic, magnifying bins with few measurements.

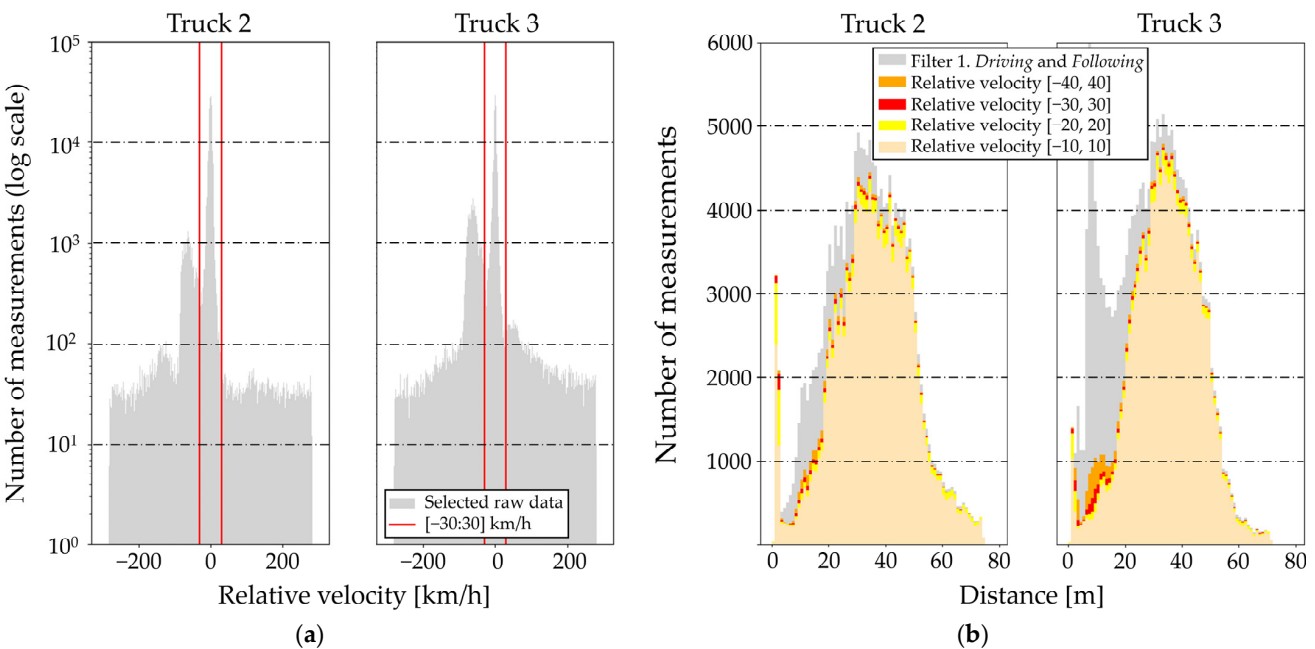

**Figure 4.** (**a**) Relative velocity histograms. (**b**) Distance histograms with relative velocity filters.

The histograms show two data spikes. Relative to a forward-facing sensor mounted in a moving truck, stationary surroundings have negative speed comparable to the speed of the truck. Assuming traffic moves at the speed limit, oncoming vehicles are measured with negative relative velocities at twice the speed limit. Similarly, measurements from the preceding truck have relative velocities fluctuating around 0 km/h. These two clusters of relative velocities appear as vertical spikes in Figure 4a. The cluster from the preceding truck was the largest. This is as expected in car-following situations, which necessitate continuous acceleration and deceleration [37]. The smaller clusters in Figure 4a had an order of magnitude fewer measurements, all of which had relative velocities in the $-30$ to $-160$ km/h range. They included static objects, oncoming traffic and measurement noise. Thus, for the relative velocity filter (filter 2), choosing $-30$ km/h as the lower threshold was natural, placing it at the local minimum between the spikes. Similarly, setting the upper threshold at $+30$ km/h made it so the entire top spike was included, while minimizing the inclusion of measurements from the noise floor.

As shown in Figure 4b, the radar data were also subjected to different relative velocity filters, starting at $\pm 40$ km/h, successively constricting by $\pm 10$ km/h steps until the narrowest filter of $\pm 10$ km/h. The color of each filter reflects remaining data points *after* that filter has been applied. For instance, remaining data after relative velocity filtering at $\pm 30$ km/h are shown in red. The $\pm 40$ km/h filter left a spike from 10–20 m for truck 3. The $\pm 30$ km/h filter removed most of the spike, and subsequent filter constriction did not cause notable differences. Thus, $\pm 30$ km/h was chosen, striking a balance between retaining most measurements corresponding to the preceding trucks and minimizing unwanted ones (oncoming and stationary objects), while including situations with sudden braking and acceleration in the platoon, which are perhaps the most interesting ones from a safety and fuel savings perspective. The radar sensors were listed as having a velocity accuracy of $\pm 0.25$ m per second. At $\pm 30$ km/h cut-offs, this amounts to a possible deviation of 3%, which is considered acceptable.

Still, as shown in Figure 4b, relative velocity filtering did not remove the leftmost spikes of measurements at distances too short to represent the preceding truck (0–5 m). Since the platoon drove with 2–3 s gaps, these do not represent the preceding truck. Calculations using Equation (A1), based on the 30° vertical field-of-view of the sensor, show that the road can be detected at 8–8.5 m forwards, meaning further away than the spikes. Thus, they are most likely clutter due to roadside detections, such as such as rock faces, tunnel walls, guardrails and signposts. Compared to the large, reflective rear walls of preceding trucks, such measurements should presumably be noisy, i.e., have small SNR values. Conversely, if these measurements did originate from the preceding truck, they should have had large accompanying SNR values. SNR values at short distances were indeed found to be small, and filtering for SNR < 15 dB was successful in removing them. As shown in Figure 5a, 15 dB filtering fell at a local minimum or saddle point between two distinct SNR data spikes. The data were also subjected to different SNR filters, illustrated in Figure 5b, but the filter was not constricted further, as doing so caused removal of data points with distance values around 40 m, likely corresponding to the preceding truck.

Since the radar sensors were able to measure up to 5 detections simultaneously, the next filter (filter 4) involved selecting only one desired target in multi-target instances: The one most likely corresponding to the preceding truck. The distance value of each detection was compared with the average distance values of the previous 10 measurements (moving average). The detection with the smallest difference was chosen. However, occasional single-object detections had distances which were quite different from the general trend. In such cases, the algorithm had no choice but to select the only detection available. This produced spikes or drops in distance which affected the moving average. This problem was subsequently minimized by filter 5, which downsampled the data to 1 Hz (one measurement per second) by averaging all distance measurements within each second. This temporal resolution was considered sufficient, and also allowed for direct coupling between curated radar data and event durations from video footage.

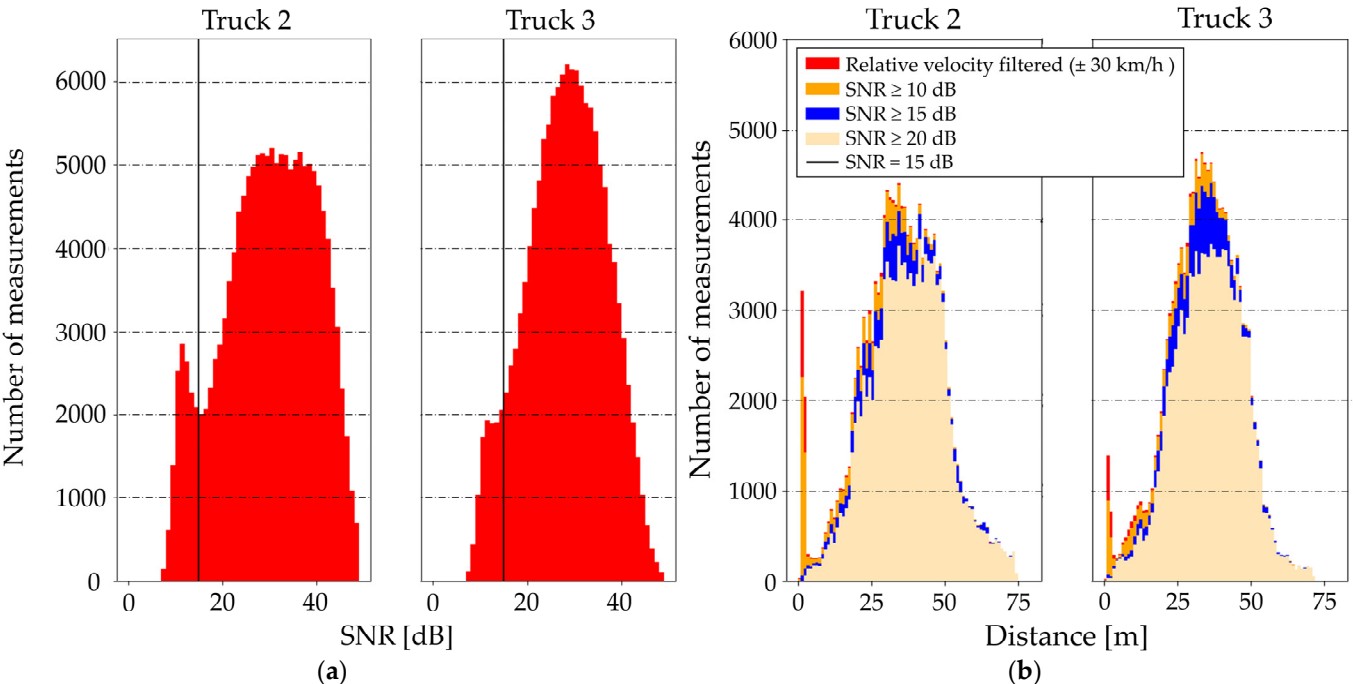

**Figure 5.** (**a**) SNR histograms after relative velocity filtering. (**b**) Distance histograms with different SNR threshold (after relative velocity filtering).

Inspecting plots of distance versus time for downsampled radar data revealed the presence of sporadic periods entirely void of points, and also to periods when points were scattered (i.e., having varying distance values following no obvious trend). To understand these detections, video footage was coded for *LWRR* and *LBRR*. Filter 6 used these codes to include only data collected in *LWRR* periods, and to exclude data collected during *LBRR* periods. Cameras malfunctioned at times, during which *LWRR* and *LBRR* could not be coded. Associated radar data were discarded, ensuring methodological consistency.

In brief, filters 1 and 6 were based on manual video codes, while filters 2 and 3 were based on recorded radar metrics, making them the most interesting ones in terms of radar operation. Filters 4 and 5 were computational heuristics. See Appendix A.3 for more details.

## 3. Results and Discussion

This section explores effects of the filtering process, before discussing differences between expected and empirical maximum distance ranges. It also explores the ability of the sensors in measuring the preceding truck in different driving situations. Finally, suggestions for future work are made. See Appendix A.4 for complete data tables.

### 3.1. Impacts of Filtering

Impacts on the number of data points are detailed, before discussing the effects on recorded metrics: relative velocity, signal-to-noise ratio and distance.

Figure 6 shows the sizes of datasets as a function of the filtering steps. The datasets of trucks 2 and 3 were affected similarly. Downsampling (filter 5) included a mean of six data points and a mode of seven. Figure 7 is a distance histogram showing the effects of all filtering steps. Note how early filters mostly remove measurements at close distances.

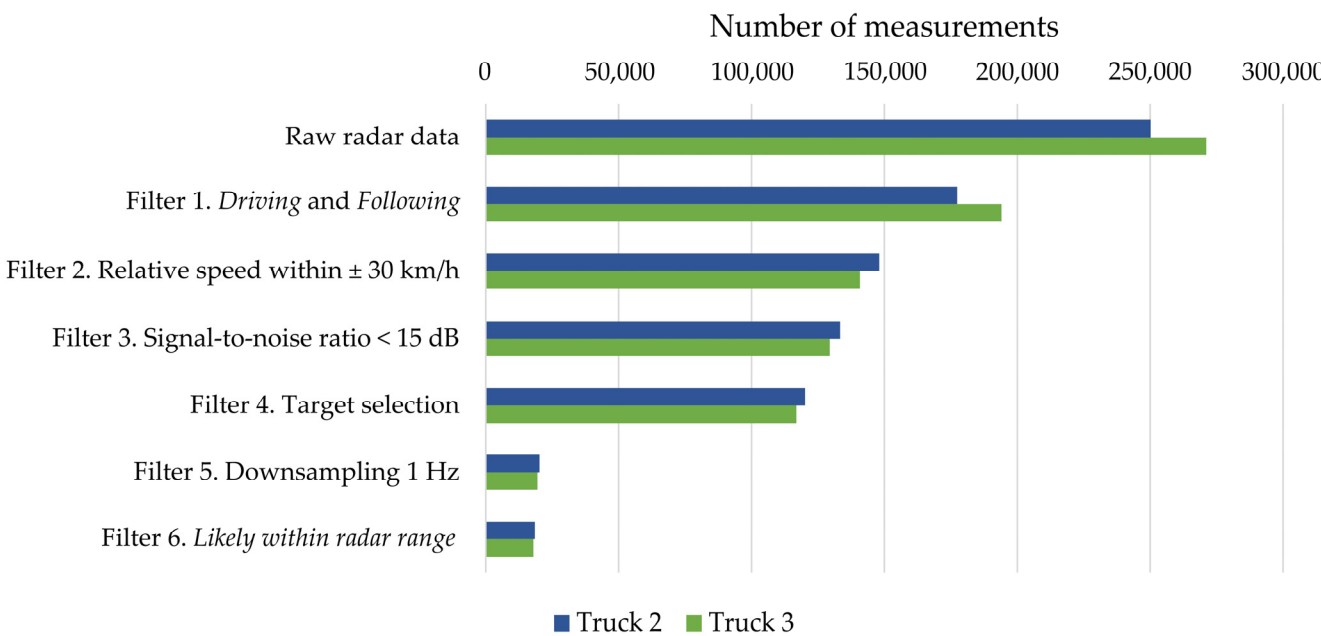

**Figure 6.** Number of measurements after each filtering step has been applied.

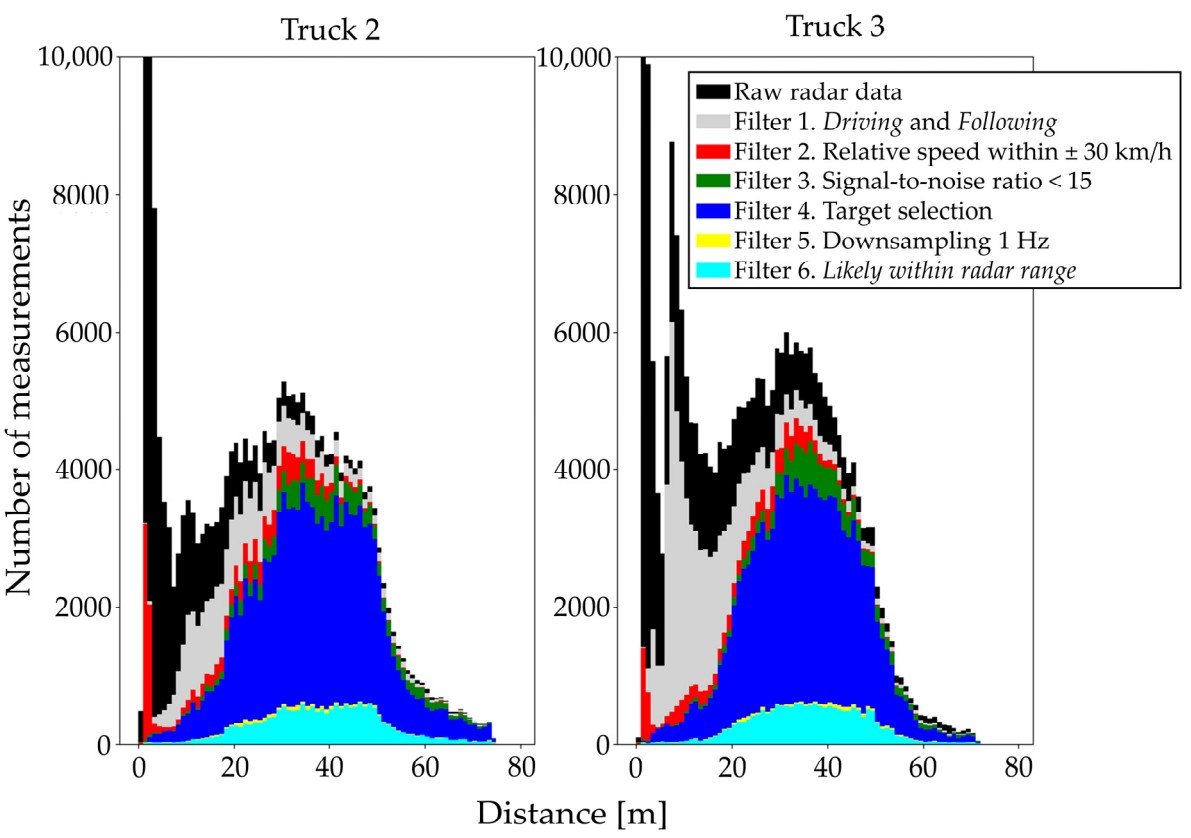

**Figure 7.** Histogram of measured distances as a function of filtering steps.

3.1.1. Relative Velocity

Overall, filtering reduced relative velocity data spread. Still, filter 1 increased relative velocity and data spread for trucks 2 and 3. In breaks, trucks were parked behind one another. Thus, removal of data from break periods serves to increase data spread, since these metrics differed more in driving periods. Excluding filter 1 (*Driving* and *Following*), filter 2 (relative velocity ± 30 km/h) affects the measured relative velocity the most, by

removing data points corresponding to oncoming vehicles. This filter also reduced the average relative velocity to approximately $-0.5$ km/h for both trucks. Thus, while average relative velocities approach zero (as they should in car-following situations), they remain slightly negative. The negative value is due to most detected objects (with the exception of the preceding trucks) heading toward the sensors (as opposed to receding from them). The curated datasets for both trucks had slightly more instances of negative than positive relative velocities. The impact of filter 2 is greater for truck 3 than for truck 2, despite both having approximately equal sizes of datasets. It removes 20% of measurements from truck 3 and only 12% from truck 2, versus the number of measurements remaining after filter 1. In fact, the dataset of truck 3 enters filter 2 with a larger average relative velocity, presumably due to platoon instability and weight differences in which perturbations caused harsh braking for truck 3, which would naturally tend to occur at short distances, which filter 2 ended up removing (cf. Figure 7). Subsequent filtering steps slightly reduce the variability of relative velocity measurements, suggesting that erroneous detections are gradually removed.

### 3.1.2. Signal-to Noise Ratio (SNR)

Filtering caused average SNR to stabilize around 29–32 dB for all trucks. Overall, filtering decreased SNR data spread. The minimum SNR was only affected by filter 3 (SNR < 15 dB). As intended, filter 3 subjected the data to a step-change, starting at 6.7–6.9 dB and ending up for all three trucks at 15.1 dB. The effects of each filter gradually diminish. Interestingly, all trucks measured different maximum SNR values, which were reduced in filter 5 (downsampling). As shown in Figure 5b, SNR filtering worked as intended by removing measurements at short distances.

### 3.1.3. Distance

Distance was the only recorded metric which was not used as a basis for filtering. Average inter-vehicle distance values, shown in Table 3, suggest that truck 3 drove closer to its preceding truck than what truck 2 did. This was visually confirmed from video footage. After curation, average values were 38.6 and 36.1 m, for trucks 2 and 3, respectively. Still, distributions of distances appear to differ somewhat, with spikes at 0–10 m for truck 2, and 10–20 m for truck 3.

**Table 3.** Distance metrics (in meters) after each filtering step.

| Filtering Step | Truck Number | | | | | |
| | 2 | | | 3 | | |
| | Average | Maximum | Standard Deviation | Average | Maximum | Standard Deviation |
|---|---|---|---|---|---|---|
| Raw | 26.5 | 74.4 | 17.6 | 25.2 | 71.5 | 15.9 |
| 1 | 33.5 | 74.4 | 14.8 | 29.3 | 71.5 | 14.5 |
| 2 | 35.5 | 74.4 | 14.3 | 34.4 | 71.5 | 12.3 |
| 3 | 37.4 | 74.4 | 13.0 | 35.5 | 71.5 | 11.5 |
| 4 | 37.2 | 74.4 | 12.9 | 35.3 | 71.5 | 11.5 |
| 5 | 38.2 | 74.3 | 13.0 | 36.1 | 71.4 | 11.5 |
| 6 | 38.6 | 74.3 | 12.9 | 36.1 | 71.4 | 11.3 |

Unfiltered maximum distance values for trucks 2 and 3 were 74.4 and 71.5 m, respectively, and were virtually unaffected by filtering. Only filter 5 (downsampling) reduced maximum distances, and only by 0.1 m in both cases. This resulted in curated maximum distances of 74.3 and 71.4 m. While distance ranges were never systematically tested, maximum distance values may be used as a proxy. While driving on straight road segments, inter-vehicle distances oscillated. The trucks would occasionally drive closely together before becoming dispersed, travelling with spacing between the trucks so large that each preceding truck was eventually located beyond sensor range. Thus, for each truck, measure-

ments should exist at the radar range boundary. As stated, filter 6 (*LWRR*) did not further reduce maximum distance values than what filter 5 did. Figure 7 shows that filter 6 did not further reduce maximum distance values since it had little effect beyond approximately 50–55 m. Since *LWRR* video coding was carried out visually, somewhat imprecisely, the farthest radar measurements still tend to appear in *LWRR*-filtered data. Had it been more precise, maximum distance values following filter 5 would best represent *actual* upper sensor ranges.

The aforementioned maximum distance values for trucks 2 and 3 fall short of the theoretical ranges 75.0 and 73.1 m by 0.9% and 2.4%, respectively. Hence, both sensors appear to underperform slightly versus expected ranges, but still fall within the estimate provided by the supplier. The deviation is smallest for truck 2. Since the datasets from both trucks are otherwise comparable, the chosen parameter values $N_s$ and $f_0$ for the radar sensor in truck 2 appears to be preferable. Future testing could explore this.

For the curated radar data, average relative velocity and average SNR were inspected as a function of inter-vehicle distances. For both trucks 2 and 3, these metrics were calculated within successive 10-m distance bins. The lower bin was 0–10 m, and the upper bin was 70 m and above. For all bins, average relative velocities fell within the error margin of the sensors. On the other hand, average SNR values are more interesting. For all bins, average SNR very seldomly fell below 30 dB, and remained high even at long distances, indicating that the radar cross-sections of the preceding trucks are sufficiently large to allow for longer detection ranges, cf. Tables A8 and A9 in Appendix A.4. Thus, parameters could likely have been chosen to achieve a maximum range approaching, and perhaps even exceeding, 100 m. The radar user manual illustrates that cars can be detected at 75 m, while buildings can be detected at 100 m. Due to the size of the truck rear walls, they may provide 'building-like' detection ranges. This study tracked large, reflective metal back walls of semi-trailers, with cross-sectional areas exceeding 8 square meters. The back walls of semi-trailers are larger than those of other vehicles and presumably facilitate higher-quality detections. Measuring distances to smaller vehicles should be tested. While the maximum distance range may be greater when detecting trucks than passenger cars, situations may exist where trucks are less favorable. For instance, when traversing sharp curves at short distances, the radar cross-section changes as truck back walls change angles in relation to the sensors, as opposed to being located perpendicular to them. Some military ships and aircraft are deliberately made from planar surfaces joined at sharp angles to achieve radar stealth. Similarly, truck back walls are also two-dimensional planar surfaces. In curves, reflected signals may scatter away from the receivers and cause data loss. This may also occur if the preceding truck is located outside the main lobe of the radar antenna. Such diffraction effects were reported in [38].

### 3.2. Radar Sensor Operation in Different Driving Situations

The curated radar data were coupled with video codes to determine whether the sensors captured high-quality inter-vehicle distance measurements when they should have been able to. Filter 6 aimed to remove periods when respective preceding trucks were located beyond radar range. We now check whether the radars were able to account for the remaining duration. *Driving* is explored, and also *Tunnel* and *Roundabout* subcategories.

Outputs from filter 6 contain radar data at 1 Hz, so aggregated video code durations (in seconds), when filtered for *LRWW*, are directly comparable to curated radar data. The proportions are shown in Table 4. Both sensors retained a similar number of measurements in each condition. As Table 4 shows, the sensors detected the preceding truck in most situations where it had the opportunity to do so. Using *Tunnels* to illustrate: Trucks 2 and 3 drove in *Tunnels* while having their respective preceding trucks within radar range (*LWRR*) for approximately 26 min (1647 and 1688 s, respectively). Aggregated over these same periods, their radar sensors outputted 1576 and 1609 curated measurements (at 1 Hz) after filter 6. As proportions, this yields 96% and 95%, respectively. Examples from each condition are shown in Table A5 in Appendix A.2.

**Table 4.** Proportions of curated radar data retained vs. *LWRR*-filtered aggregated event durations.

| Condition | Trucks | |
|---|---|---|
| | **2** | **3** |
| *Driving \** | 85% | 83% |
| *Tunnels* | 96% | 95% |
| *Roundabouts* | 88% | 89% |

*\* Driving includes Tunnels and Roundabouts.*

Radar data and video codes can be used to visualize and explore excerpts of curated radar data (*LWRR*) from different driving intervals. Inter-vehicle distances are plotted versus time. Excerpts indicate large variability in inter-vehicle distances as a function of infrastructure type and road standard, as suggested in [23].

Since *Tunnel* and *Roundabout* video codes have non-zero durations, a choice had to be made regarding from which truck to visualize them. Truck 2 was chosen, being the middle truck in the platoon for 88% of the drive. Note that, at the time resolution used (1 min divisions), the difference would have been negligible if visualizing codes from one of the other two vehicles. Oncoming trucks seemed to cause the platoon to slow down, which also affected inter-vehicle distances, particularly on narrow roads and in sharp curves. Therefore, *Oncoming truck* video codes are shown. Being point events, these could be visualized for each truck separately. The horizontal axes (d hh:mm) were not fixed, so excerpts have slightly different durations (between 5 to 10 min). As the field study took place over two days, d-values are either 1 or 2.

3.2.1. Visual Verification of Maximum Range

Figure 8 shows three excerpts from *Driving*. While no tunnels or roundabouts are shown, statistics from this category in Table 4 also include those durations, thus showcasing diverse, complex driving segments. While serving as followers for 7 h, trucks 2 and 3 drove with their respective preceding trucks within radar range for approximately 6 h. All excerpts in Figure 8 had the preceding truck within range, so trends for measured distances appear mostly continuous. The top excerpt stems from an old, narrow road section without centerlines. Trucks often adjusted their speeds, including when encountering opposing trucks, as revealed by reduced inter-vehicle distances. The middle excerpt stems from the traversal of a flat and wide high-quality road with a 90 km/h speed limit and gentle horizontal curves. Here, the trucks maintained constant distances over long time periods, and opposing trucks did not influence the platoon. The individual data points from truck 2 which are located below the general trend were chosen by the algorithm since the sensor did not detect other data points during that logging instance. In general, truck 2 had more such erroneous measurements than truck 3 did, but it is unclear why this is the case. The lower excerpt illustrates the descent of a challenging mountain pass (negative 6% gradient), where trucks 2 and 3 reduced their speed repeatedly, to avoid speeding and becoming located too close to their respective preceding truck. These excerpts suggest that truck platooning is more suitable on wide, modern roads than on old roads with adverse horizontal and vertical alignment.

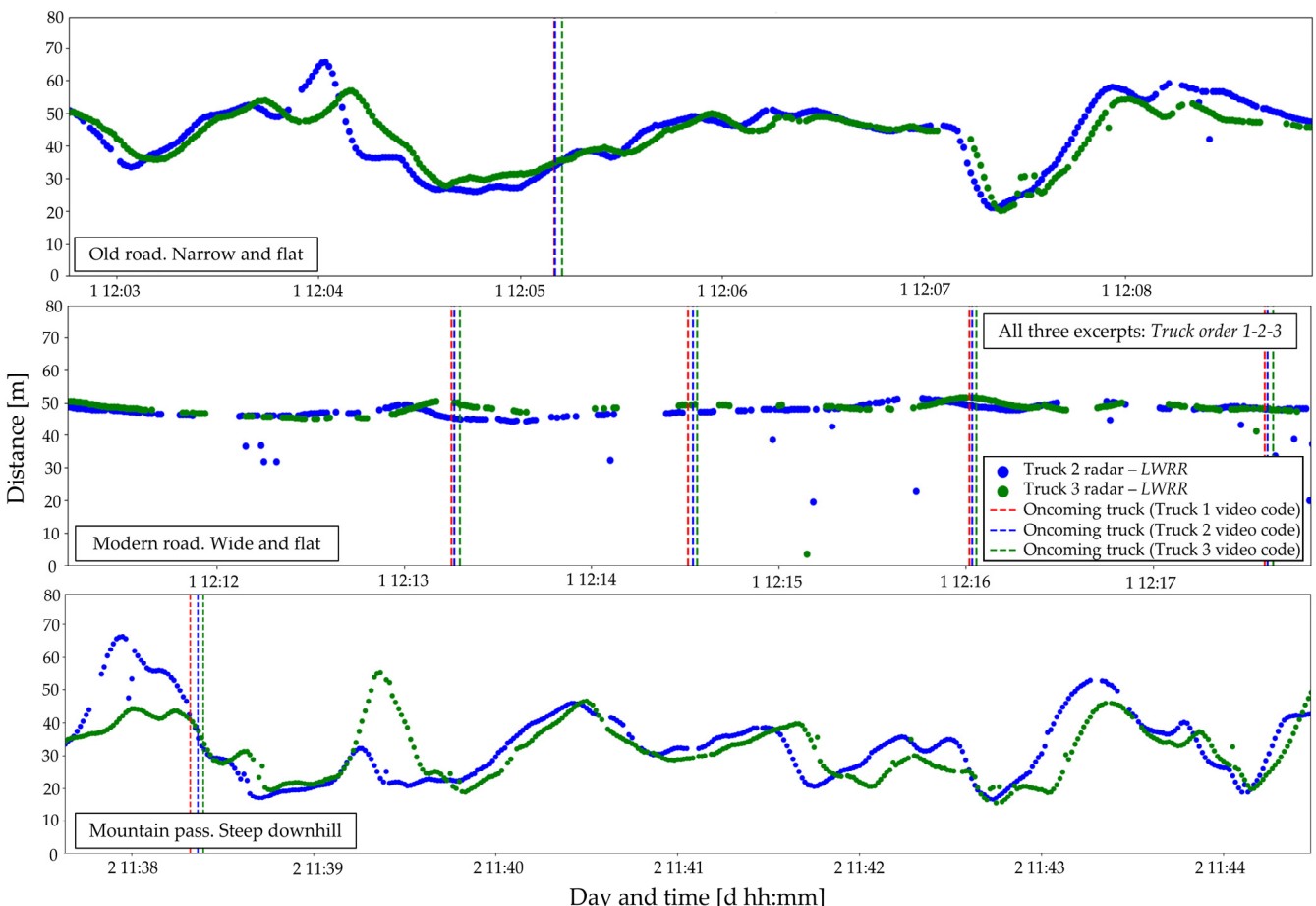

**Figure 8.** Inter-vehicle distances measured from three separate driving excerpts, *LWRR*, during which radar range was never exceeded.

Figure 9 shows three excerpts during which, for both trucks 2 and 3, the inter-vehicle distance appears to have occasionally exceeded maximum radar ranges. Blue (truck 2) and green (truck 3) horizontal lines illustrate the farthest distances detected, which differ somewhat between excerpts. Nonetheless, for radar sensors in trucks 2 and 3, respectively, the maximum distances are approximately 75 and 70 m, which support the aforementioned statistics-based radar range estimates. The three excerpts stem from two different mountain passes. The two upper excerpts correspond to traversal of Mountain pass A, with very difficult combinations of sharp horizontal curves and vertical gradients. Videos were also useful in inspecting the radar data after curation. The combination of long inter-vehicle distances and horizontal curves occasionally caused the preceding truck to be obscured by rock walls at the inner part of right-turn curves (pale shading). Mountain pass B, shown in the lowermost excerpt, was more forgiving in terms of road alignment.

In the upper excerpt, the data from truck 3 (green) at timestamp 16:56 potentially reveals the presence of a phenomenon which, together with the high average SNR values measured at large distances, suggests that sensors are capable of measuring the preceding truck far away. It appears as if measurements which naturally belong to the top of the green curve are folded down, instead of occurring at 90–100 m, where extrapolation would place them. Thus, it looks like the radar in truck 3 does measure the preceding truck, despite it being located beyond the maximum range imposed by the chosen parameters. Stated otherwise, the radar appears to measure points beyond its unambiguous range, referring to the maximum distance a target can have while it can be guaranteed that the reflected pulse from that target corresponds to the most recent transmitted pulse [14,39]. At this timestamp, it appears as if the returned signal is associated with the wrong transmitted pulse, so

the range becomes ambiguous. Filtering for *LWRR* and *LBRR* served to remove most durations where such long-distance samples may have been folded into the ambiguous range. However, the manual video coding process did not remove all such instances. The presence of folding may have influenced distance metrics to the downside.

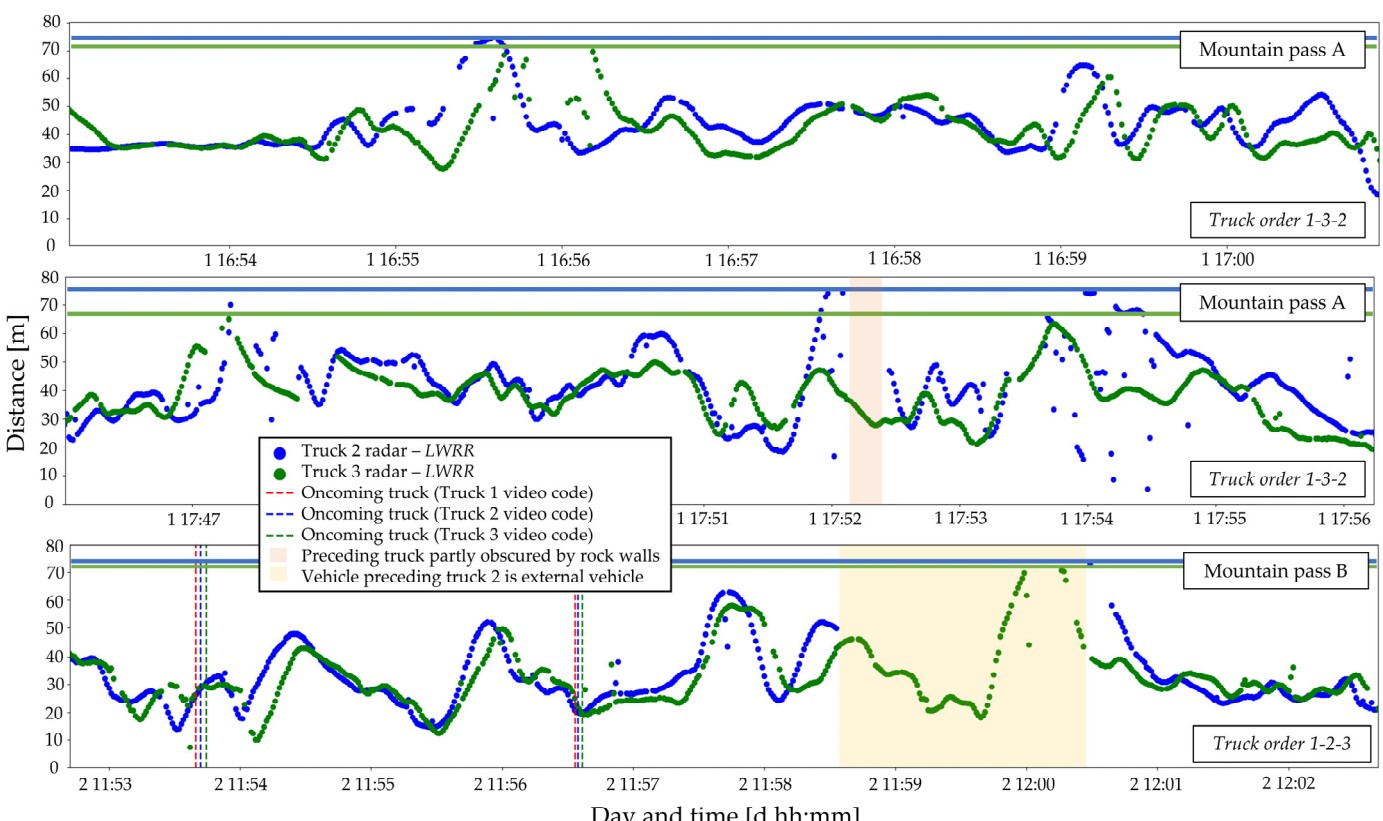

**Figure 9.** Inter-vehicle distances measured from three separate driving excerpts, *LWRR*, during which radar range was exceeded. Horizontal lines show farthest measurements.

Data from truck 2 in the middle excerpt of Figure 9 are noisier than for all other excerpts shown, but it is unclear whether the folding phenomenon occurs here. The outage for truck 2 in the lower excerpt stems from a period when a passenger car partly overtook the platoon, being sandwiched between truck 1 and 2 until also overtaking the lead truck. Data from such periods were removed by filter 1 (*Driving* and *Following*).

### 3.2.2. Tunnels

It is unclear how tunnels affect the ability of the sensors to measure inter-vehicle distances. Tunnels may reduce operational complexity, as rock walls cause peripheral narrowing of roadside areas. However, walls, lighting and ventilation elements may introduce clutter. Such features are less frequent on roads in natural terrain.

All *Tunnel* driving occurred in truck order *1–2–3*. Figure 10 shows excerpts from six representative *Tunnel* traversals (green shading). Driving periods outside tunnels have white backdrops. All excerpts were coded as *LWRR*, except for the period between the two tunnels in the top excerpt, when data were lacking for truck 2. This period was coded as *LBRR*, as truck 1 was located far away. Comparing Figures 8 and 10, it seems as if filter 4 (target selection) chooses erroneous data points at comparable frequencies both inside and outside tunnels. Thus, tunnel driving does not appear to degrade radar operating conditions. In tunnels, maximum distance values for trucks 2 and 3 were reduced by 8–9%, and the distance standard deviation dropped by 18% and 15%, respectively. Thus, inter-vehicle distances were moderated by *Tunnels*, causing closer, more uniform driving at lower speeds. This made preceding trucks occupy a larger part of radar the field-

of-view. However, measurements collected in tunnels were noisier than those collected during *Driving*, indicated by lower SNR values (weaker signal). For both followers, *Tunnel* filtering reduced mean SNR by 3%, maximum values by 6–8% and standard deviations by 7–11%. This did not affect the curated radar data when plotted: In Figure 9, erroneously selected targets in tunnels appear to have distance values at similar deviations to the trend, compared to erroneously selected targets outside tunnels. Thus, it appears as if inter-vehicle distance measurements between platooning trucks are not adversely affected by tunnels.

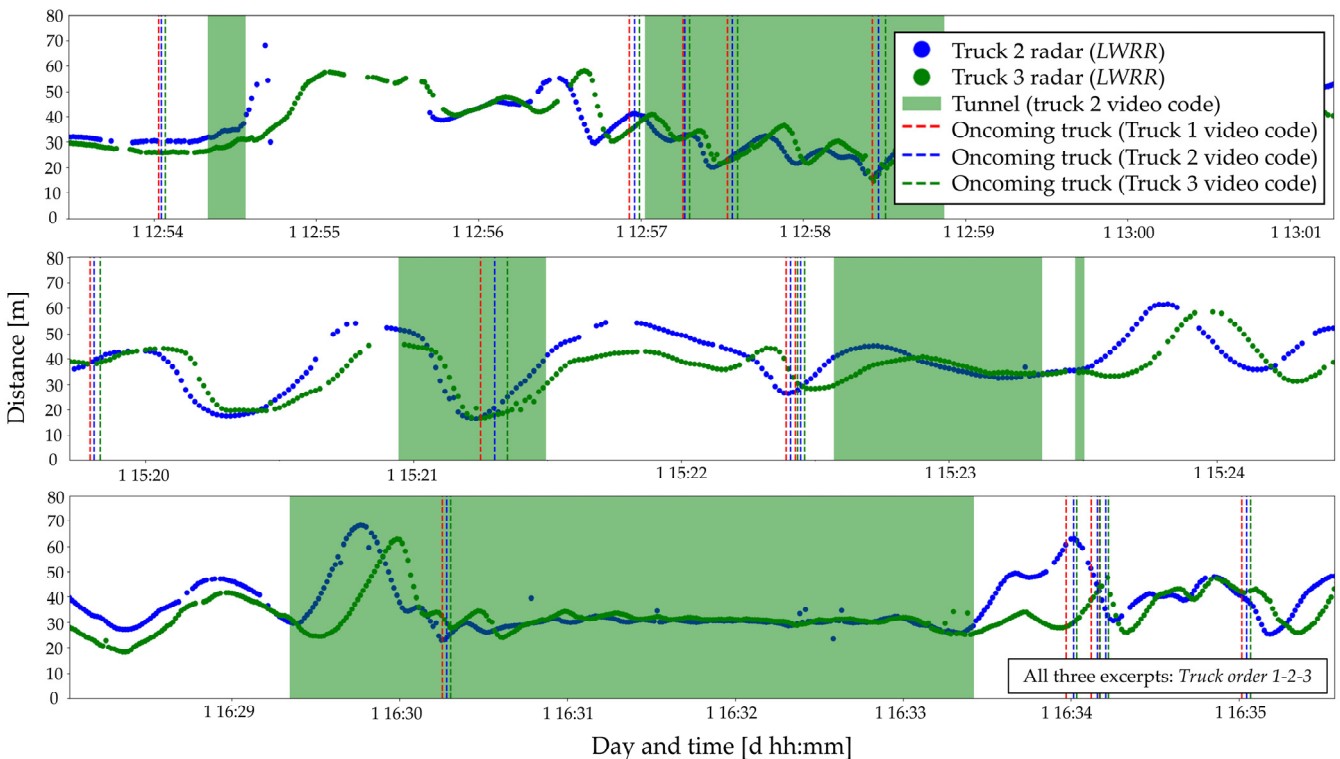

**Figure 10.** Radar data for three separate truck driving excerpts, *Tunnel* and *LWRR*.

### 3.2.3. Roundabouts

Roundabouts may allow for exploring the effects of sharp road curvature on radar operation. Most roundabout traversals involved straight movements, encountering little to no traffic. Figure 11 shows excerpts from five traversals (pale red shading), all of which occurred in truck order *1–2–3*. Driving periods outside roundabouts have white backdrops. The first traversal in the middle excerpt involved trucks 1 and 2 performing a full revolution to get rid of external vehicles located between trucks 2 and 3. Truck 3 had *Other vehicles* preceding it before entering this roundabout, so its data were removed by filter 1. The traversal in the lower excerpt involved all three trucks revolving one round. Trucks 2 and 3 both had 10 traversals as followers, with 5 straight, 1 left turn and 4 right turns. For both trucks, only half of the aggregated *Roundabout* durations had the preceding truck within field-of-view, and even when accounting for field-of-view, 11–12% of data are lost. Figure 11 shows that data points retained in roundabouts generally have scattered distance values which are too large to represent the preceding truck. Sharp curve radii and limited antenna beam width resulted in lost field-of-view to the preceding truck, so radar sensors detected irrelevant objects until field-of-view was regained after the turn.

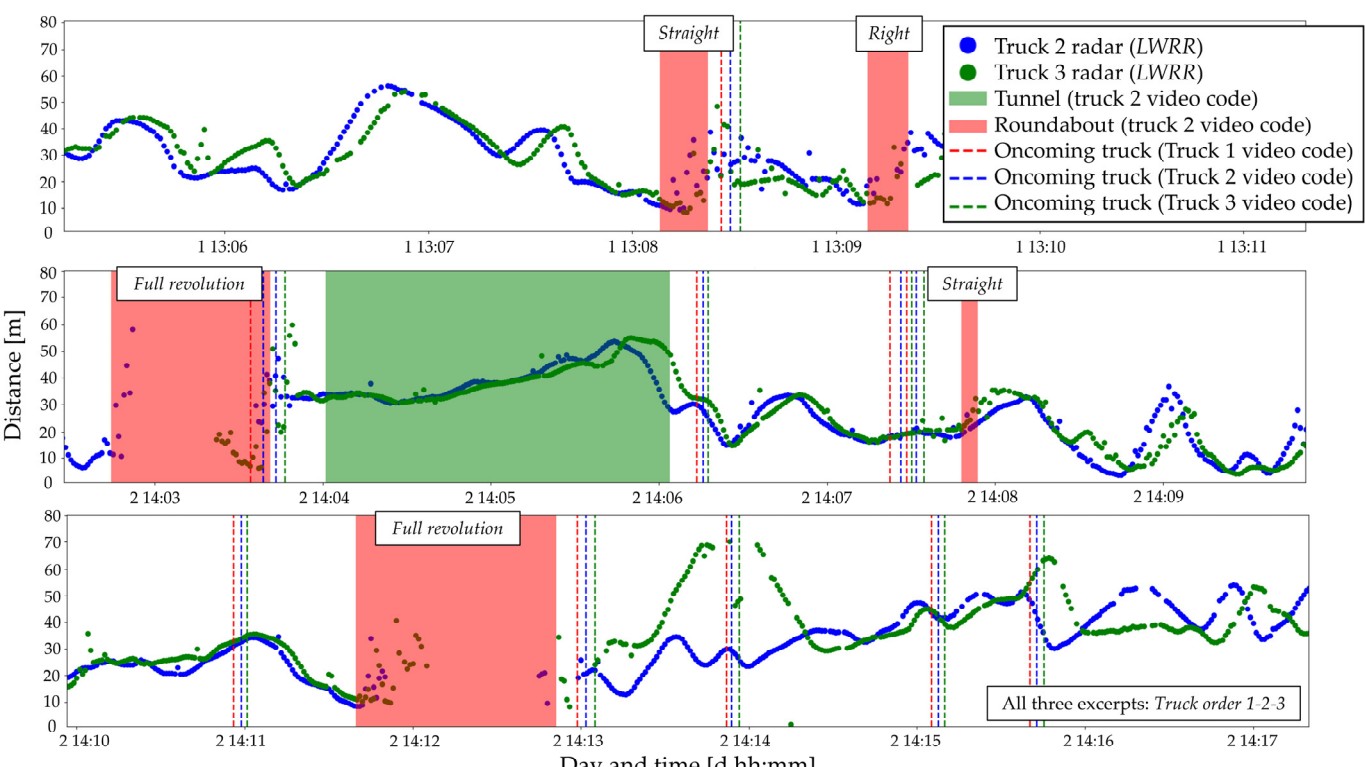

**Figure 11.** Radar data from three separate driving excerpts with *Roundabouts*, *LWRR*.

Average SNR values were 30 and 32% lower for trucks 2 and 3 in *Roundabouts* than for *Driving*, respectively, so the radar sensors detected noisier data. Proportions of radar data as a function of *LWRR*-filtered *Roundabout* durations (88–89%), are greater than for *Driving* (83–85%). This is likely due to the trucks being grouped closer together when traversing *Roundabouts*. Speeds were also lower, causing smaller inter-vehicle distances and greater spatial concentration of measurements. For trucks 2 and 3, mean distances were 66–62% shorter during *Roundabouts* than during *Driving*. The exploration shows that preceding trucks are tracked poorly when the platoon passes through roundabouts.

### 3.3. Suggestions for Future Work

This study explored the extent to which uRAD radar sensors could capture inter-vehicle distances during truck platooning. Several suggestions have been identified.

First, mode 4 (7 Hz) might be more suitable than mode 3 (13 Hz). As data were downsampled anyway, the trade-off between higher update rate (mode 3) and enhanced properties for ghost target reductions in complex scenarios (mode 4) should be explored.

To simplify data collection, radars could be remotely engaged from escort vehicles. If field trials are undertaken in areas with adequate cell coverage, virtual network computing (VNC) could be used to remote control all Raspberry Pi microprocessors. Radar data may also be visualized and coded in real-time. Future work may validate the radar data against GNSS positions, if undertaken in areas with good conditions for GNSS receivers. The performance of the sensors should also be compared against a known baseline, i.e., radars with known characteristics, such as those listed in [14], in controlled environments.

The cross-sectional signature of the trucks was not measured, and scattering effects were not explored. If present, such effects would be reflected in SNR values if systematically inspected at equal distances while varying the angle of the back wall of the preceding truck, resembling the set-up in [40]. Windshield attenuation effects were also not studied.

Synchronization of videos and radar data, and the subsequent process of video coding, worked well. However, both should preferably be automated, to reduce post-processing

efforts and related human errors. Traffic and infrastructure events may be identified directly from radar data. Herein, tunnels and roundabouts were coded from the moment where each respective truck entered them, and to the moment when each respective truck left them. Later, video codes were overlaid on the radar data collected from each truck. However, since radar data shows the *preceding truck*, this may have introduced a systematic error. Perhaps video codes from the preceding truck should have been used instead. For tunnels, this is not particularly problematic since tunnel traversals (with *LWRR*) had long durations (on average 1.1 min). Thus, entering and leaving the tunnel occupies a very small part of the total duration. For roundabouts, however, the preceding truck had often traversed $\frac{1}{4}$–$\frac{1}{2}$ of the roundabout before the truck in question entered it, and it was first coded. Distinctions may be made between the preceding truck being located beyond range, and it being located laterally beyond field-of-view, as these are different phenomena. All data points may also be given metadata for all relevant video codes, simplifying video inspections of interesting events in the data.

Statistical approaches may allow cut-off values for filters 2 and 3 to be chosen automatically. In the target detection step, distance filtering could also be considered, perhaps discarding data points with distances deviating significantly from the general trend. This may solve the problem of erroneous single-object detections. Established data filtering, target tracking [41] or clustering techniques [42–44] could also be used, alongside more computationally complex methods for annotating or labelling combinations of radar and camera footage, e.g., in [31,45], and perhaps also machine learning approaches [46].

At times, dashboard cameras malfunctioned due to power issues, totaling 6% of driving time, during which 10% of all radar data were logged. As it was not possible to determine whether the sensors had reasonable operating conditions in these periods, the data were discarded. Mitigations include redundant cameras and independent power supplies.

## 4. Conclusions

Anteral uRAD radar sensors for Raspberry Pi were tested for estimating inter-vehicle distances between trucks. Three trucks participated in a real-word platooning field study. Data from integrated sensors were unavailable. Comparable results were found from the sensors in the two rearmost trucks, suggesting that they are feasible for this use case. Data filtering involved a multi-faceted methodology. While also filtering based on relative velocity and signal-to-noise ratio, video footage allowed for removal of data from irrelevant periods, and for exploring sensor operation in roundabouts and tunnels. This would not have been possible without video footage. The curated radar data can be used to model expected fuel savings from truck platooning on specific types of roads and road features.

Sensor ranges were estimated at 74 and 71 m, i.e., slightly shorter than suggested by theoretical calculations. The sensors captured the preceding truck for 83–85% of the time when it was located within radar range. In tunnels specifically, 95–96% of driving time was accounted for, likely due to closer driving. Average SNR decreased 3% in tunnels, compared to all driving, but this did not appear to affect the accuracy of the target detection step. When turning in roundabouts, the field-of-view to the preceding truck was often lost, and the sensors detected their surroundings until field-of-view was regained after completing the turn, causing average SNR values to drop (30–32% lower). Straight movements in roundabouts were less problematic, as field-of-view was mostly retained.

High SNR values were observed at far distances, indicating that the sensors, with optimal parameters, may be capable of measuring preceding trucks further away. The findings suggest that simple, inexpensive radar sensors and action cameras can facilitate collection of inter-vehicle distance data from truck platooning field trials.

**Author Contributions:** Conceptualization, M.M.L.; methodology, M.M.L., T.T. (Thomas Thoresen), M.H.R.E.; software, T.T. (Thomas Thoresen); validation, T.T. (Thomas Thoresen); formal analysis, M.M.L.; investigation, M.M.L.; data curation, T.T. (Thomas Thoresen); writing—original draft preparation, M.M.L.; writing—review and editing, M.M.L., M.H.R.E., T.T. (Trude Tørset), T.L.; visualization, T.T. (Thomas Thoresen); supervision, T.T. (Trude Tørset), T.L.; project management, M.M.L.; funding acquisition, T.L. All authors have read and agreed to the published version of the manuscript.

**Funding:** This research was funded in part by the Norwegian University of Science and Technology (NTNU) and the Norwegian Public Roads Administration (NPRA), through the Innovation and Implementation initiative (D11351). The field study itself, titled "Smart and Connected Truck Train" was financed by the NPRA, cf. the Norwegian database for public procurement: https://doffin.no/nn/Notice/Details/2020-301190.

**Institutional Review Board Statement:** Not applicable.

**Informed Consent Statement:** Informed consent was obtained from all participants involved.

**Data Availability Statement:** Data are available on request from the corresponding author.

**Acknowledgments:** The main author would like to acknowledge the Anteral supplier team for answering inquiries.

**Conflicts of Interest:** The authors declare no conflict of interest.

## Appendix A

The appendix is subdivided into four parts, providing background information and methodological details for reproducibility, alongside tables with results.

*Appendix A.1. The Radar Sensors*

Anteral uRAD radar sensors, version 1.1, were used alongside Raspberry Pi 4, model B with 4 GB RAM. Micro-SD cards (16 GB) were purchased, with the Raspberry Pi operating system and uRAD software pre-installed. Technical support was also purchased. Table A1 provides an overview of the radar configuration parameters. Testing prior to the field trial is outlined in Figure A1 and Table A2. Collected test data and corresponding camera footage were sent to the supplier, who proposed recommendations for future testing. Table A3 details the parameters which were used in the field study.

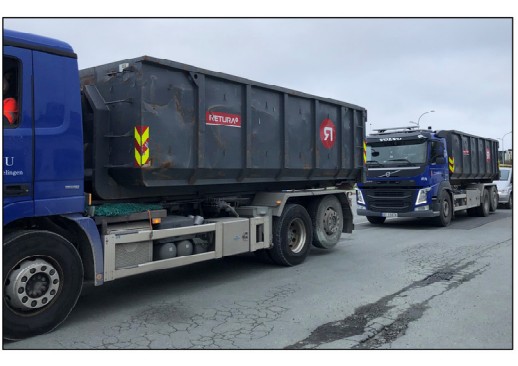
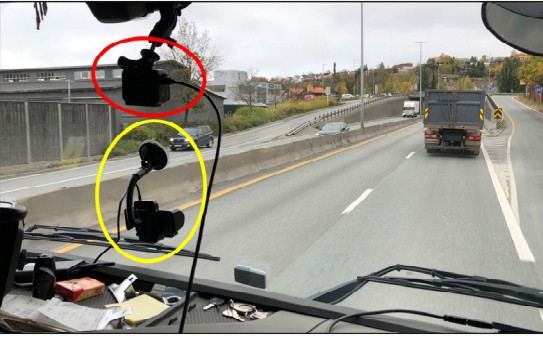
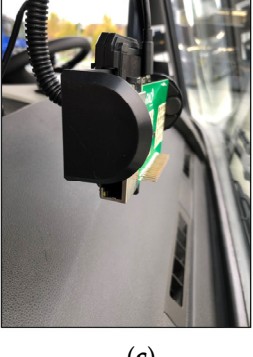

(**a**)                  (**b**)                  (**c**)

**Figure A1.** (**a**) Dump truck test set-up; (**b**) Back wall of preceding truck seen from cabin during driving, with radar sensor (yellow) and dashcam (red) circled; (**c**) Radar sensor side view.

The supplier detailed the pre-processing steps taken before sensors store data to memory: The radar transceiver chip receives the reflected signal. The mixer mixes the received (RX) signal with the transmitted (TX) signal, and outputs in-phase (I) and quadrature (Q) components at Intermediate Frequency (IF). These two analog I/Q IF signals go through a low-pass filter, an amplifier stage and a high-pass filter. Filter values are proprietary. Subsequently, the analog signal is digitalized with an ADC at 25 kHz in mode 1 and 200 kHz in

modes 2, 3 and 4. In the digital domain, the complex signal is formed, FFT obtained, and from it, range, velocity and SNR of detected targets are derived. The supplier also stated that calibration by the user is not needed, as performance is controlled in the lab prior to shipment, by measuring a constant distance of 1.5 m in mode 2.

The combination of one Raspberry Pi attached to one uRAD radar board is here termed a *device*. The device was fixed to the interior of the windshield using a universal phone suction mount with a flexible arm, oriented such that the USB-C and micro-HDMI ports pointed directly upwards (cf. Figure A1). Since Raspberry Pi microprocessors were powered on and off using a USB-C cable, this orientation facilitated easy access and line of sight from above, for cable insertion and removal. It also made sure that the sensor did not detect the mount itself, being located outside radar field-of-view. Powerbanks and USB-C cables powered the devices at optimum voltage and amperage, while making data logging independent of truck power systems and status. This eliminated potential issues with undervolting and voltage spikes from in-vehicle outlets, while leaving flexibility for when logging start and stop had to be administered. Each Raspberry Pi had a Bluetooth USB dongle for a wireless keyboard and mouse, minimizing direct device contact. The dongle added an additional reason for ensuring stable power supply. The radar supplier confirmed that the dongle radio frequency would not affect radar operation. Along with the dongle, the micro-HDMI and power cables remained plugged into the device throughout the field trial. This allowed the devices to remain vehicle-mounted throughout both days, requiring only insertion and removal of the far-ends of the cables into the screen and powerbank for interfacing with the devices and powering them devices on and off, respectively. Radar output files never exceeded 5 MB, i.e., they were unproblematic with respect to SD card storage capacity. A battery powered portable monitor was used.

With 30° vertical fields-of-view, 15° swept down from horizontal, so vertical sweep became 75°, assuming sensors were mounted perfectly level. If trucks, with uRAD antennas at height $h_a$, traversed a constant gradient road section, the road would be detected at a distance, *d*, given Equation (A1):

$$d = h_a \cdot \tan(75°) \tag{A1}$$

The presence of aftermarket dashboard tabletop surfaces required placing sensors at slightly different heights in each truck during the field study. With radar antennas at heights of 2.29 m (truck 2) and 2.15 m (truck 3), the road would be detected 8–8.5 m forwards. Having 30° fields-of-view also in the horizontal direction, radars also saw this far sideways at road level.

**Table A1.** Overview and discussion of uRAD radar parameters.

| Parameter | Discussion |
|---|---|
| Mode | Of four modes available, only modes 3 (triangular) and 4 (dual-rate) measured both distance and velocity. Velocity would enable filtering away stationary and oncoming objects, to be left with desired inter-vehicle distance to the preceding truck. Modes 3 and 4 differed in upper distance range and update rate. Mode 3 had an upper distance range of 100 m, versus 75 m for mode 4. The supplier stated that the range would also depend on the target, meaning its radar cross-section: "(…) a person is detected up to 40 m. (…) a truck, that is bigger and reflects more, (…) will be detected [at] 70 m but probably (…) much farther. 100 m is not a limitation of the radar, [but] a guide (…) for very big targets." Mode 4 should reduce ghost target detections in multi-target scenarios, at the expense of reduced range. |

**Table A1.** *Cont.*

| Parameter | Discussion |
|---|---|
| Ramp start freq., $f_0$. Operation bandwidth, $BW$ | Ramp start frequency, $f_0$, could be set as 5–195 for modes 2–4. Operation bandwidth, $BW$, meaning the frequency sweep used in modes 2–4, depends on $f_0$, and should be maximized, subject to Equation (A2), to increase accuracy and to distinguish closely located targets. For each radar, different values were chosen to avoid interference.<br><br>$$BW_{max} = 245 - f_0 \qquad (A2)$$<br><br>The $f_0$ parameter denotes the starting frequency of the waves emitted by the sensor. The sensor operates at a frequency bandwidth of 24.005–24.245 GHz, and $f_0$ values are defined (in MHz) to set the offset from the lower threshold. |
| Samples and ramp duration, $N_s$ | $N_s$ is the number of samples taken from the reflected wave to calculate distance and velocity. Highest update rate requires lowest possible $N_s$. However, a trade-off is needed, since $BW$ and $N_s$ determine maximum range, through Equation (A3).<br><br>$$Distance_{max} = 75 \cdot \frac{Ns}{BW} \qquad (A3)$$<br><br>The $N_s$ parameter serves two purposes. Firstly, it defines the duration of each wave ramp, and secondly, it outlines the sampling rate from the reflected wave, per ramp duration, which can be used to calculate output metrics. |
| Max. detected targets, $N_{tar}$ | $N_{tar}$ is the number of targets that the sensor detects, 5 being maximum. If detecting more objects, the sensor logs data for those 5 with highest SNR. $N_{tar}$ was maximized, capturing most data and providing possibility for filtering unwanted objects later. |
| Maximum detection distance, $R_{max}$ | For modes 2–4, $R_{max}$ is the maximum distance below which targets will be detected. $R_{max}$ artificially reduces the zone of interest, excluding targets beyond this distance, even if they have higher SNR than those within it. $R_{max}$ was chosen as 100 for all sensors, as this would search targets within the entire range. When asked if the sensors would stay fixed on the preceding truck in horizontal curves, the supplier stated that manual antenna modification could double the horizontal FOV, to the detriment of upper detection range. No manual modifications were made. For vertical curves, the supplier cited that the road in front of the truck, which would be more visible in vertical sag curves, could reflect the signal, masking the preceding truck. |
| Moving target indi-cator, $MTI$. Movement detection, $Mth$ | Moving target indicator ($MTI$) allowed for including data only from objects with motion relative to the sensor. $Mth$ is only relevant when using uRAD as a movement detector, and was not used. |

**Table A2.** Pre-trial testing of radar parameter configurations.

| Pre-Test Steps | Parameters | User Experience | Supplier Modifications and Recommendations |
|---|---|---|---|
| 1: Passenger car test with one radar sensor and standard graphical user interface (GUI) | Mode = 2<br>$f_0$ = 45<br>$BW$ = 200<br>$N_s$ = 200<br>$N_{tar}$ = 5<br>$R_{max}$ = 100<br>$MTI$ = 1<br>$Mth$ = 1 | • Preceding traffic recorded well.<br>• Data written to the same file each time subsequent data collection is stopped and started.<br>• Data are only written to file upon logging stop. Susceptible to data loss if equipment malfunctions. Cannot distinguish driving segments.<br>• Epoch time format impractical and not human-readable. | • Switch off Mth, as it is not relevant for the application.<br>• Replaced GUI with Python script for increased update rate.<br>• Code rewritten to create new output files upon each logging start.<br>• Output files are now named with "start logging time" in human-readable format.<br>• Data are now continuously written to file during logging, as opposed to batch writing upon logging termination. |

**Table A2.** *Cont.*

| Pre-Test Steps | Parameters | User Experience | Supplier Modifications and Recommendations |
|---|---|---|---|
| 2: Test with two dump trucks. Follower with dashcam and one radar sensor  | Mode = 2, 3<br>$f_0 = 45$<br>$BW = 200$<br>$N_s = 200$<br>$N_{tar} = 5$<br>$R_{max} = 100$<br>$MTI = 1$<br>$Mth = 0$ | • The preceding truck was recorded well, except in curves and intersections.<br>• Mode 3 is preferred over mode 2; makes it easy to filter away stationary objects and oncoming vehicles based on relative velocity.<br>• Filtering distance values for relative velocity exceeding $\pm 20$ km/h removes much noise. | • Field study involves three sensors, not one.<br>• Finalized parameters were recommended to avoid interference, yet maximize sensor range. |

**Table A3.** Radar parameter configurations.

| Truck ID | Common Parameters | Specific Parameters |
|---|---|---|
| 2 | Mode = 3<br>BW = 200 MHz<br>$N_{tar}$ = 5 targets<br>$R_{max}$ = 100 m<br>MTI = 1 (active)<br>Mth = 0 (inactive) | $f_0$ = 5 MHz<br>$N_s$ = 200 samples |
| 3 | | $f_0$ = 25 MHz<br>$N_s$ = 195 samples |

*Appendix A.2. Video Footage, Video Synchronization and Manual Video Coding*

Low-resolution video (LRV) file segments were converted to the MP4 format and merged using free Bandicut software [47]. Merged videos were imported to BORIS version 7.12.2. LRV files were used, since original MP4 files were too large for BORIS to handle. LRV files were 864 by 480 pixels, while original files were 1920 by 1080 pixels, both with 60 fps frame rates. Conversion reduced the file size by an order of magnitude, while retaining sufficient video quality for coding. Table A4 shows two code definitions, while examples of state events are shown in Table A5.

**Table A4.** Examples of video code definitions.

| Video Code | Definition |
|---|---|
| *Driving* (S) | *Driving* starts when the truck is fully inside the correct lane on the roadway, with the steering wheel turned straight. It stops just before the driver starts turning the wheel, with the intention of entering driveways, parking areas or stop pockets. Except during *Break* and periods of camera malfunctions, every other video code is coded only when *Driving* is also active. |
| *Break* (S) | All time that is not *Driving*, is defined as *Break*. This includes maneuvering in and out of driveways, parking areas and stop pockets. |

**Table A5.** Examples of coded state events for radar data filtering.

| Event | Illustrations |
|---|---|
| *Tunnel* | 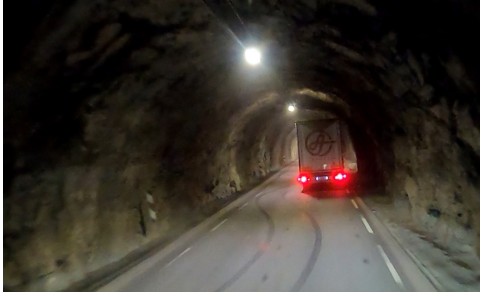 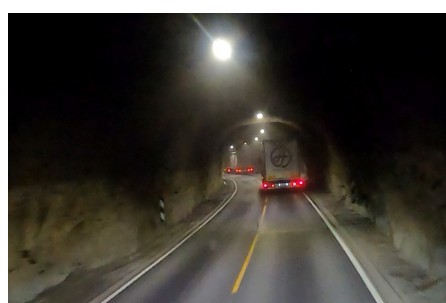 |
| *Roundabout* (Left): Straight; (Right) Right-turn | 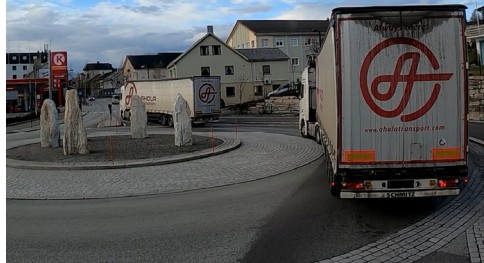 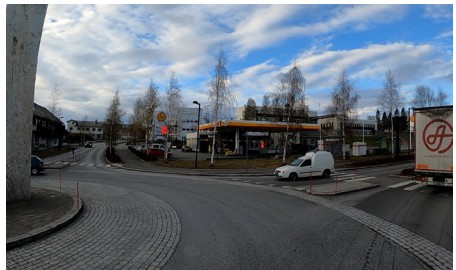 |
| *LWRR (likely within radar range)* | 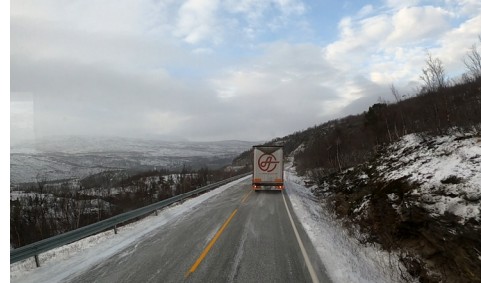 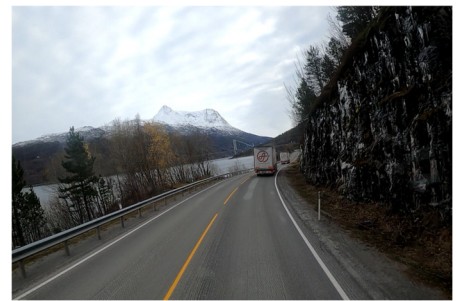 |
| *LBRR (likely beyond radar range)* | 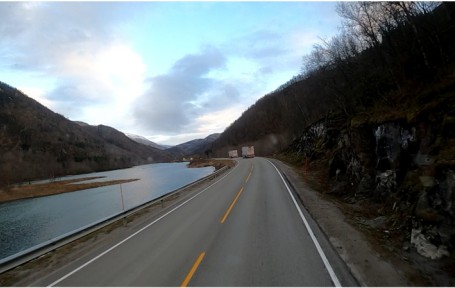 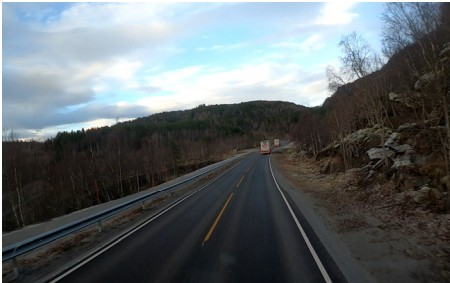 |

*Appendix A.3. Radar Data Curation*

Radar data post-processing was carried out in Python 3.10. The Python libraries Pandas, NumPy, datetime, Matplotlib and openpyxl were used. All radar data were extracted into a Pandas DataFrame. In multi-target scenarios, objects were placed successively within the DataFrame, by descending SNR. The DataFrame contained the following data columns: Time {*datetime*}, Distance {*float*}, Velocity {*float*}, SNR {*float*} and Object number {*int*}. Since radar timestamps did not correspond to local time, datetime shifts were calculated based on the previously corrected *date and time* of each *Radar logging* instance, and the *date and time* with which the radar output files were named (cf. Table A2). Datetime shifts were added to the Time column, correcting all measurements. All radar data were merged into one DataFrame and saved as a Pickle file prior to curation.

Filters 4 (target section) and 5 (downsampling) are illustrated in Figures A2 and A3.

Figure A2 shows a 7-min period for truck 2. Chosen targets are blue, discarded ones purple, and the moving average turquoise. Averaging across 10 loggings, instead of fewer, reduced noise in turbulent situations. Inspection of radar data, e.g., when traversing *Roundabouts*, showed that approximately 10 loggings were needed after the video code ended, for distance to the preceding truck to stabilize.

The presence of lone blue data points located away from the blue trend in Figure A2, shows that this algorithm may choose the wrong target. Erroneous selections are those data points which clearly suppress the moving average distance value. Figure A3 shows a 10-min excerpt of downsampled radar data (blue), alongside data after filter 1 (gray).

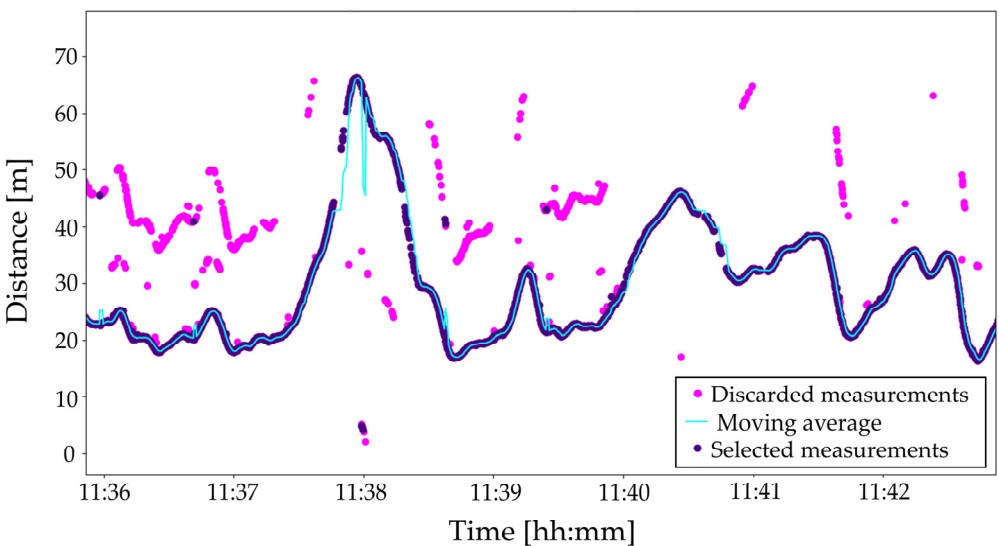

**Figure A2.** Truck 2 excerpt. Target selection when radar measures multiple objects simultaneously.

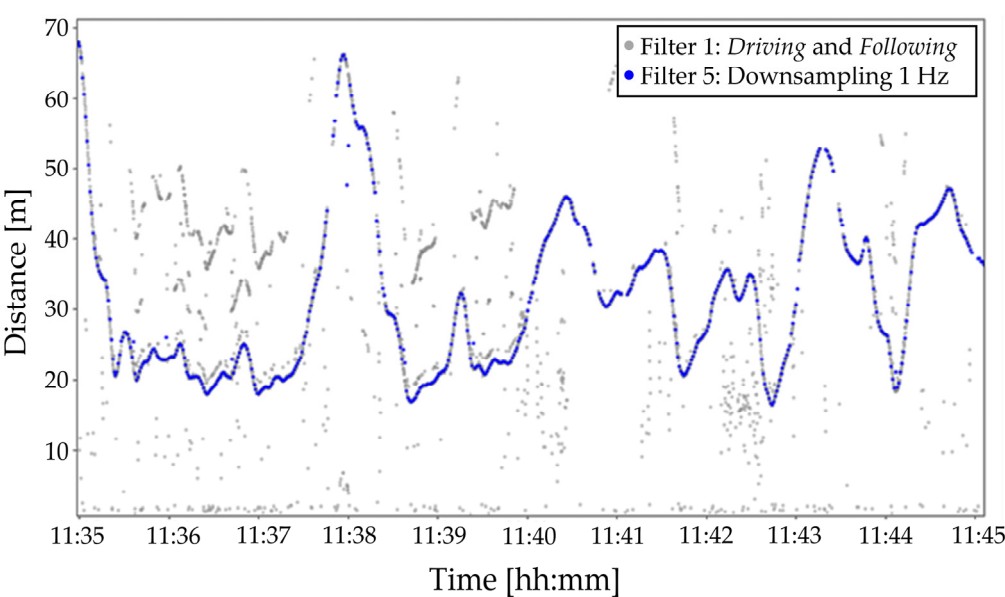

**Figure A3.** Truck 2 excerpt following filters 1 and 5.

*Appendix A.4. Results and Discussion*

Tables A6–A13 show statistics from each filtering step, the analysis of relative speed and SNR as a function of distance bins, and radar operation in different driving situations.

**Table A6.** Relative velocity statistics (km/h) for trucks 2 and 3 after each filtering step.

| Filtering Step | Truck Number | | | | | | | |
| --- | --- | --- | --- | --- | --- | --- | --- | --- |
| | 2 | | | | 3 | | | |
| | Min | Avg | Max | Std | Min | Avg | Max | Std |
| Raw | −280.9 | −5.7 | 281.0 | 40.4 | −280.7 | −9.4 | 280.7 | 45.9 |
| 1 | −280.9 | −7.7 | 281.0 | 43.7 | −280.7 | −11.3 | 280.7 | 48.9 |
| 2 | −29.9 | −0.5 | 29.9 | 5.5 | −29.9 | −0.5 | 29.9 | 5.5 |
| 3 | −29.9 | −0.3 | 29.9 | 4.8 | −29.9 | −0.3 | 29.7 | 4.6 |
| 4 | −29.9 | −0.2 | 29.9 | 4.4 | −29.9 | −0.2 | 29.7 | 4.2 |
| 5 | −29.6 | −0.2 | 28.6 | 4.3 | −29.9 | −0.2 | 27.2 | 4.0 |
| 6 | −29.6 | −0.1 | 28.6 | 4.2 | −29.7 | −0.1 | 27.2 | 3.8 |

**Table A7.** SNR statistics (dB) for trucks 2 and 3 after each filtering step.

| Filtering Step | Truck Number | | | | | | | |
| --- | --- | --- | --- | --- | --- | --- | --- | --- |
| | 2 | | | | 3 | | | |
| | Min | Avg | Max | Std | Min | Avg | Max | Std |
| Raw | 6.8 | 24.6 | 53.9 | 10.8 | 6.6 | 22.6 | 51.4 | 9.7 |
| 1 | 6.9 | 27.6 | 53.9 | 10.6 | 6.7 | 24.2 | 51.4 | 9.9 |
| 2 | 6.9 | 29.9 | 53.9 | 9.9 | 6.8 | 28.1 | 51.4 | 8.5 |
| 3 | 15.1 | 31.8 | 53.9 | 8.3 | 15.1 | 29.5 | 51.4 | 7.3 |
| 4 | 15.1 | 32.9 | 53.9 | 7.9 | 15.1 | 30.4 | 51.4 | 7.0 |
| 5 | 15.1 | 31.6 | 49.7 | 6.9 | 15.1 | 29.4 | 47.7 | 5.9 |
| 6 | 15.1 | 31.8 | 49.7 | 6.9 | 15.1 | 29.5 | 47.7 | 5.8 |

**Table A8.** Average relative velocity (km/h) and SNR (dB) for truck 2 as a function of distance.

| Distance Bins | Avg. Relative Velocity (km/h) | Average SNR (dB) | # Measurements | % of Total |
| --- | --- | --- | --- | --- |
| 0–10 | −0.4 | 30.0 | 204 | 1% |
| 10–20 | −1.0 | 32.3 | 1191 | 6% |
| 20–30 | −0.3 | 31.1 | 3327 | 18% |
| 30–40 | −0.1 | 31.8 | 5074 | 27% |
| 40–50 | 0.0 | 32.4 | 5519 | 30% |
| 50–60 | 0.2 | 31.4 | 2151 | 12% |
| 60–70 | −0.1 | 31.2 | 813 | 4% |
| 70+ | 0.7 | 32.2 | 191 | 1% |

**Table A9.** Average relative velocity (km/h) and SNR (dB) for truck 3 as a function of distance.

| Distance Bins | Avg. Relative Velocity (km/h) | Average SNR (dB) | # Measurements | % of Total |
| --- | --- | --- | --- | --- |
| 0–10 | −0.3 | 29.4 | 199 | 1% |
| 10–20 | −1.2 | 29.1 | 1113 | 6% |
| 20–30 | −0.4 | 28.6 | 4173 | 23% |
| 30–40 | 0.0 | 29.0 | 5642 | 31% |
| 40–50 | 0.1 | 30.2 | 4988 | 28% |
| 50–60 | 0.0 | 31.0 | 1502 | 8% |
| 60–70 | 0.9 | 31.6 | 270 | 2% |
| 70+ | 0.4 | 31.4 | 29 | 0% |

**Table A10.** Relative velocity statistics (km/h) for trucks 2 and 3 during *LWRR* and *Tunnel*.

| Truck | Relative Velocity | | | |
|---|---|---|---|---|
| | **Avg** | **Min** | **Max** | **Std** |
| 2 | 0.1 | −19.7 | 28.6 | 3.5 |
| 3 | 0.1 | −19.3 | 18.8 | 3.3 |

**Table A11.** Distance (meters) and SNR (dB) statistics for trucks 2 and 3 during *LWRR* and *Tunnel*.

| Truck | Distance | | | | SNR | | | |
|---|---|---|---|---|---|---|---|---|
| | **Avg** | **Min** | **Max** | **Std** | **Avg** | **Min** | **Max** | **Std** |
| 2 | 36.4 | 10.6 | 68.2 | 10.5 | 30.8 | 15.1 | 46.9 | 6.4 |
| 3 | 36.2 | 13.8 | 65.0 | 9.6 | 28.5 | 15.7 | 44.0 | 5.2 |

**Table A12.** Distance (meters) and SNR (dB) statistics for trucks 2 and 3 during *LWRR* and *Roundabout*.

| Truck | Distance | | | | SNR | | | |
|---|---|---|---|---|---|---|---|---|
| | **Avg** | **Min** | **Max** | **Std** | **Avg** | **Min** | **Max** | **Std** |
| 2 | 23.3 | 8.6 | 58.4 | 10.1 | 24.5 | 16.2 | 39.5 | 5.2 |
| 3 | 22.4 | 6.3 | 60.0 | 11.2 | 22.4 | 15.1 | 41.5 | 4.4 |

**Table A13.** Relative velocity statistics (km/h) for trucks 2 and 3 during *LWRR* and *Roundabout*.

| Truck | Relative Velocity | | | |
|---|---|---|---|---|
| | **Avg** | **Min** | **Max** | **Std** |
| 2 | −6.5 | −27.8 | 11.2 | 9.1 |
| 3 | −5.9 | −25.7 | 10.5 | 8.9 |

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
