# Peer review of "Using Low-Cost Radar Sensors and Action Cameras to Measure Inter-Vehicle Distances in Real-World Truck Platooning"

_asi, doi:10.3390/asi6030055_

Round 1

Reviewer 1 Report

Dear authors, the paper presents a very interesting research but I think the main problem is the paper organization. 

The paper is more of a technical report than a research paper. Suggest indicating this in the title or introduction. 

The paper is difficult to read and most part of the interesting work is reported in the appendixes and which makes it very difficult to understand the work. My suggestion is to reorganize the paper including the most salient information in the main text to facilitate the reading. 

Also, the most part of interesting results are in the appendixes, so I suggest including these in the discussion. This will improve the presentation and the quality of the work. 

Maybe the authors an try to be more concise and provide only useful information.  

I suggest also to better explaining the choice of a 24 GHz radar. The technology is evolved to 77 GHz radars so this choice must be better justified.  

The paper has a good English style.

Reviewer 2 Report

1. This manuscript describes an interesting sensor application technology. It is written in great detail, but I hope the authors will try to follow the style of the academic paper instead of the experimental report.

2. The length of the article can be significantly reduced, and redundant content should be deleted as much as possible, otherwise readers may feel that the logic is not clear enough.

3. Please explain as much as possible the motivation and reasons for the technical methods, and clarify the innovative points compared to existing literatures.

4. It is best to add some theoretical analysis on the performance of radar sensor.

Round 2

Reviewer 1 Report

The paper is now more clear and simpler to read.